# RGMem: Renormalization Group–inspired Memory Evolution for Language Agents

**Ao Tian** [1,2]  **Yunfeng Lu** [1,2]  **Xinxin Fan** [3]  **Changhao Wang** [4,5]  **Lanzhi Zhou** [4,5]  **Yeyao Zhang** [6]  **Yanfang Liu** [4,5]

## Abstract

Personalized and continuous interactions are critical for LLM-based conversational agents, yet finite context windows and static parametric memory hinder the modeling of long-term, cross-session user states. Existing approaches, including retrieval-augmented generation and explicit memory systems, primarily operate at the fact level, making it difficult to distill stable preferences and deep user traits from evolving and potentially conflicting dialogues.To address this challenge, we propose RGMem, a self-evolving memory framework inspired by the renormalization group (RG) perspective on multi-scale organization and emergence. RGMem models long-term conversational memory as a multi-scale evolutionary process: episodic interactions are transformed into semantic facts and user insights, which are then progressively integrated through hierarchical coarse-graining, thresholded updates, and rescaling into a dynamically evolving user profile.By explicitly separating fast-changing evidence from slow-varying traits and enabling non-linear, phase-transition-like dynamics, RGMem enables robust personalization beyond flat retrieval or static summarization. Extensive experiments on the LOCOMO and PersonaMem benchmarks demonstrate that RGMem consistently outperforms SOTA memory systems, achieving stronger cross-session continuity and improved adaptation to evolving user preferences. Code is available at https://github.com/fenhg297/RGMem

[1]School of Reliability and Systems Engineering, Beihang University, Beijing, China [2]National Key Laboratory of Reliability and Environmental Engineering Technology [3]State Key Laboratory of AI Safety, Institute of Computing Technology, Chinese Academy of Sciences [4]School of Computer Science and Engineering, Beihang University, Beijing, China [5]State Key Laboratory of Complex & Critical Software Environment [6]LightSail, China. Correspondence to: Yanfang Liu <hannahlyf@buaa.edu.cn>.

*Proceedings of the 43rd International Conference on Machine Learning*, Seoul, South Korea. PMLR 306, 2026. Copyright 2026 by the author(s).

## 1. Introduction

Modern dialogue agents built on large language models are increasingly expected to sustain personalized interactions over extended periods, spanning multiple sessions and evolving user states (Li et al., 2025; Chhikara et al., 2025). However, long-term personalization exposes a fundamental mismatch: interaction histories grow without bound, while the model's reasoning at any moment is constrained by a finite context window (Collins et al., 2015; Xiao et al., 2024). This mismatch gives rise to a core challenge in long-term dialog personalization—how to maintain stable user representations across sessions while remaining responsive to new and potentially contradictory evidence. Existing memory and retrieval mechanisms struggle to achieve this balance. Limited context windows and the *lost-in-the-middle* effect (Liu et al., 2024; Zhong et al., 2024) make long-dialog reasoning unreliable, while static parametric memory is difficult to update incrementally (Wang & Chen, 2025). Although retrieval-augmented generation (RAG) (Edge et al., 2024; Guo et al., 2024) and explicit memory systems attempt to address this problem at the fact level, they are often dominated by lexical overlap and recency bias, making it difficult to abstract stable, higher-level traits across sessions. More fundamentally, dialog personalization is inherently multi-scale: it involves concrete events (micro), cross-situational regularities (meso), and long-term abstractions (macro), giving rise to the classic *stability–plasticity dilemma*.

From this perspective, abstraction is a selective, scale-dependent transformation that is triggered by sufficient evidence and applied to preserve both stability and adaptability.

Yet despite recent progress, existing memory systems still lack a principled account of how user profiles should evolve under continuous and often conflicting evidence.To reason about constrained memory dynamics, we draw inspiration from the RG perspective (Ma, 1973; Parisen Toldin, 2022; Tu, 2023), which studies how stable macroscopic structure emerges through iterative coarse-graining and rescaling. Rather than treating RG as a physical model, we adopt it as an *engineering lens* for organizing and evolving long-term conversational memory across abstraction scales.

Motivation: Resorting to the theory of renormalization

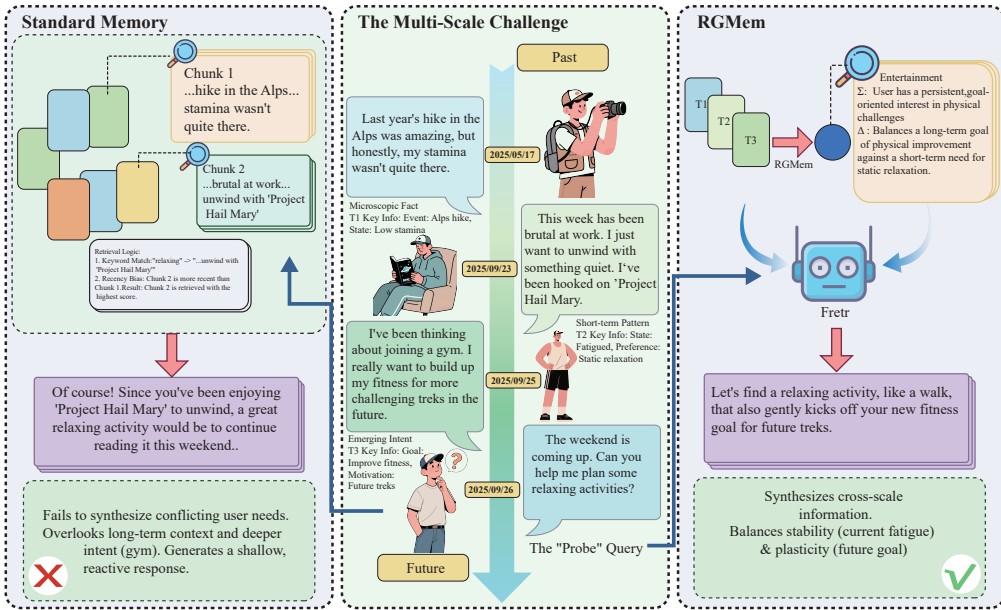

*Figure 1.* A comparative illustration of memory models encountering the multi-scale Challenge.

An illustration of how different memory models respond to the same multi-scale dialogue scenario. **Left: Flat Memory.** Standard memory systems retrieve fragmented information driven by keyword matching and recency, often missing deeper, long-term user intent. **Right: RGMem.** RGMem progressively integrates fine-grained interactions into stable, higher-level user representations while preserving important contextual conflicts, enabling balanced and proactive responses under complex, cross-scale user needs.

group, we reveals four key insights towards the predicament of long-term conversational memory for language agents:

**Insight 1. Effective Information Density is Maximized via Hierarchical Coarse-Graining.**

**Insight 2. User Profile Updates Exhibit Phase-Transition-Like Dynamics.**

**Insight 3. Separating Slow and Fast Variables Resolves the Stability–Plasticity Dilemma.**

**Insight 4. Long-Term User Profiles Exhibit Macroscopic Invariance Beyond Fact-Level Stability.**

To convince these insights, we also provide formal analytics from theoretical viewpoint in Appendix D. Motivated by these insights, we propose a novel self-evolving memory framework RGMem, it operationalizes hierarchical coarse-graining and thresholded profile evolution, multi-facet experiments demonstrate our approach can significantly improve the performance by **7.08** points on LOCOMO and **8.98** points on PersonaMem than the best baseline.

## 2. Related Work

### 2.1. Dialog Personalization and Implicit Memory

Implicit memory approaches encode user information directly into model parameters through fine-tuning or parameter-efficient adaptation (Wang et al., 2024; Tan et al.,

2025b; Wei et al., 2025; Hu et al., 2022; Yu et al., 2025). To manage the stability–plasticity dilemma within parameters during continual learning, recent advances explore decomposing update subspaces for knowledge isolation (He et al., 2026) or designing fine-grained routing in Mixture-of-Experts (MoE) to prevent catastrophic forgetting (Chen et al., 2026). These methods excel at capturing stylistic patterns and soft preferences without explicit retrieval, but require parameter updates, limiting interoperability with closed-source models and cross-agent profile sharing. Moreover, while some cutting-edge paradigms simulate cognitive intuition via continuous latent memory evolution (Zhu et al., 2026), latent or KV-cache-based memories lack fine-grained auditability, controlled updates, and rollback mechanisms, making them unsuitable for traceable, evolving user profiles. Consequently, implicit memory systems struggle to support long-term personalization under continuous and conflicting evidence.

### 2.2. Explicit Memory and Retrieval-Augmented Generation

Explicit memory systems externalize user information for persistent storage and retrieval during generation (Rasmussen et al., 2025; Chhikara et al., 2025). These approaches improve traceability and online updates, but typically operate at the fact or paragraph level, emphasizing retrieval quality rather than memory organization or evo-

lution. Without principled mechanisms for abstraction, contradiction management, or cross-scale integration, such systems often accumulate noise and inconsistencies over time (Pan et al., 2025; Tan et al., 2025a). In contrast to flat retrieval paradigms, RGMem models memory as a multi-scale evolving system, explicitly separating factual evidence from higher-level, slowly-varying user traits.(see Fig. 1)

## 2.3. Hierarchical Memory and Profile Evolution

To manage long interaction horizons, recent research has increasingly adopted hierarchical memory architectures that decouple fine-grained observations from high-level abstractions. Representative approaches include recursive summarization trees (Rezazadeh et al., 2025) and pyramidal indices (Hu et al., 2025) that organize context at varying granularities. More complex formulations utilize layered knowledge graphs to segregate episodic traces from semantic concepts, as seen in GraphRAG (Edge et al., 2024), HippoRAG (Gutiérrez et al., 2024; 2025), and Ari-Graph (Anokhin et al., 2024). While these structures enhance retrieval density, their evolution mechanisms remain static or linear: abstraction typically occurs via uniform bottom-up propagation (Li et al., 2026; Wu et al., 2025) or fixed-interval consolidation. Lacking explicit scale-dependent control over when structural reorganization should occur, these systems struggle to resolve the stability–plasticity dilemma, often over-fitting to transient noise or failing to consolidate genuine profile shifts. In contrast, RGMem models memory as a multi-scale dynamical system, where abstraction is governed by thresholded, non-linear phase transitions rather than uniform aggregation policies.

## 3. Methodology

### 3.1. Preliminaries and Motivation

The RG is a theoretical framework originally developed in statistical physics to explain how stable macroscopic structure emerges from microscopic interactions across scales (Wilson, 1983; Pelissetto & Vicari, 2002; Litim, 2001; Zinn-Justin, 2007; Yakhot & Orszag, 1986). Abstracted from its physical origins, RG can be viewed as a general principle for organizing multi-scale information: high-frequency, task-irrelevant details are progressively integrated out through coarse-graining, while scale-invariant, task-relevant structure is preserved along a renormalization flow. Similar RG-inspired ideas have appeared in machine learning, including connections to hierarchical abstraction in deep networks (Chen et al., 2018; Di Sante et al., 2022; Bény, 2013; Wang & Liu, 2025), the Information Bottleneck principle (Tishby & Zaslavsky, 2015; Mehta & Schwab, 2014), and hierarchical pooling or graph coarsening methods (Ying et al., 2018).

However, existing RG-inspired approaches mainly address static representations, whereas long-term conversational memory requires modeling how representations evolve across abstraction scales. This gives rise to a fundamental stability–plasticity dilemma: a memory system must integrate transient, fine-grained interactions while preserving stable macroscopic user traits over time. Concretely, this entails balancing internal consistency, fidelity to new evidence, and representational simplicity.

To make this intuition explicit, we introduce an *abstract design objective*, which we refer to as an *effective Hamiltonian*, to qualitatively characterize the desirability of a given user-profiling theory $\mathcal{T}$:

$$\mathcal{H}(\mathcal{T}) = \alpha E_{\text{con}}(\mathcal{T}) + \beta E_{\text{fid}}(\mathcal{T} \mid \mathcal{D}_{L0}) + \gamma E_{\text{com}}(\mathcal{T}) \tag{1}$$

where $E_{\text{con}}$ penalizes internal contradictions within the profile, $E_{\text{fid}}$ measures deviations from the accumulated microscopic evidence $\mathcal{D}_{L0}$, and $E_{\text{com}}$ discourages overly redundant or fragmented representations.

Importantly, this Hamiltonian is not meant to be explicitly optimized. Instead, it serves as a conceptual objective guiding the design of memory evolution mechanisms, since direct optimization over high-dimensional textual memory states is intractable.

Accordingly, RGMem adopts a renormalization-inspired perspective in which user profiles evolve through coarse-grained, scale-specific transformations, heuristically approximating the minimization of $\mathcal{H}(\mathcal{T})$ across abstraction levels. Guided by this perspective, RGMem is designed as a multi-scale memory system that (i) performs early coarse-graining over raw dialogue to suppress noise, (ii) updates user profiles incrementally through scale-aware transformations rather than repeated re-summarization, and (iii) allows stable representations to reorganize only when accumulated evidence crosses critical thresholds. Based on these principles, we formalize RGMem as a structured user-profiling model and present its algorithmic instantiation below.(see Fig. 2)

### 3.2. User-Profiling Modeling

Inspired by RG intuitions, we model long-term user profiling as a multi-scale representation that evolves from fine-grained conversational evidence to stable abstract traits, with explicit separation of abstraction levels and scale-dependent update mechanisms. Concretely, RGMem consists of a memory state space, a scale-dependent profile representation, and a set of transformation operators that drive profile evolution.

**Memory State Space.** The complete memory state is defined as

$$\mathcal{M} = \mathcal{D}_{L0} \times \mathcal{G}, \tag{2}$$

where $\mathcal{D}_{L0} = \{d_1, d_2, \dots\}$ denotes discrete episodic mem-

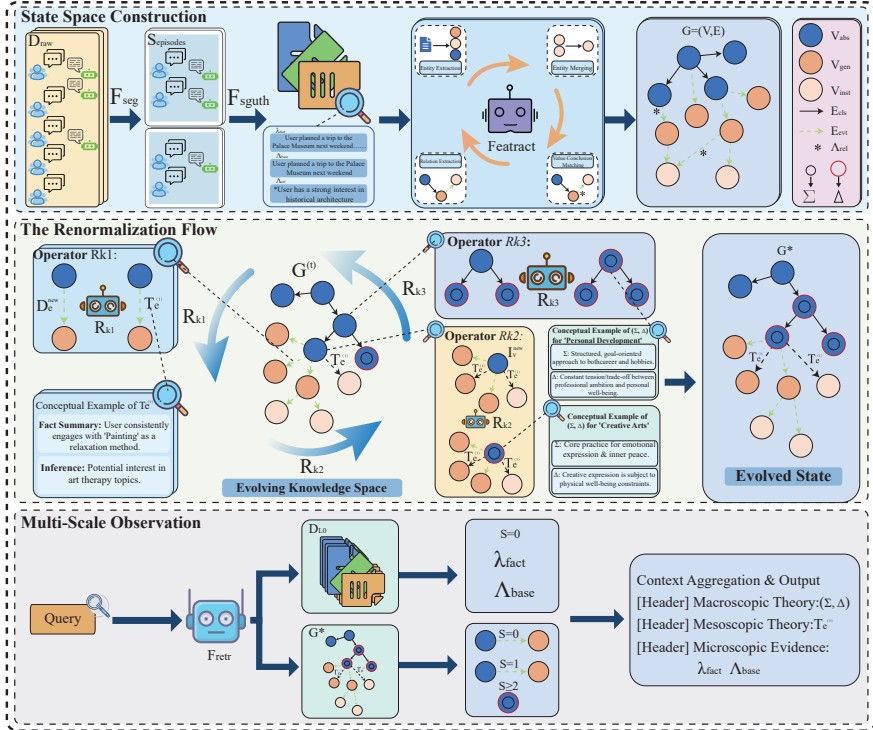

*Figure 2.* Overview of the RGMem framework. RGMem consists of three stages: (1) constructing a multi-scale memory state from raw dialogue, (2) evolving user profiles through scale-aware renormalization operators, and (3) performing multi-scale retrieval to generate coherent responses.

ory units extracted from raw dialogues, and $\mathcal{G} = (\mathcal{V}, \mathcal{E})$ is a dynamic knowledge graph with hierarchical structure . This separation distinguishes rapidly changing factual evidence from more slowly evolving relational and conceptual structures, providing the foundation for balancing stability and plasticity.

**Multi-Scale Effective Theory.** We define an *effective theory* $\mathcal{T}(\mathcal{M}, s)$ as a compact, scale-dependent representation of the user profile, parameterized by descriptors $\{\lambda_i(s)\}$. Lower scales ($s = 0$) correspond to individual episodic evidence, while higher scales ($s \geq 1$) capture aggregated patterns and abstract user traits derived from structured knowledge.

**RG Transformation.** Profile evolution across scales is modeled by a transformation operator $\mathcal{R}$:

$$\mathcal{T}^{(s+1)} = \mathcal{R}(\mathcal{T}^{(s)}). \quad (3)$$

This operator jointly performs coarse-graining and rescaling, mapping lower-level representations into abstract profile summaries. $\mathcal{R}$ is an algorithmic update rule rather than an explicit optimization procedure, governing how profiles evolve under incoming evidence.

### 3.3. Renormalization Flow: Operators and Instantiation

RGMem instantiates the multi-scale formulation in Section 3.1 through a three-layer architecture (L0–L2). L0 constructs microscopic evidence, L1 drives multi-scale memory evolution, and L2 enables scale-aware retrieval. By separating fast-changing signals from slowly evolving user traits, this design supports stable yet adaptive memory evolution. The following subsections describe each layer in detail.

#### 3.3.1. CONSTRUCTION OF MEMORY STATE SPACE

RGMem constructs its memory state space via coordinated processing in the L0 and L1 layers.

**Microscopic Evidence Space $\mathcal{D}_{L0}$.** Raw dialogue streams $D_{\text{raw}}$ are transformed into a set of microscopic memory units $\mathcal{D}_{L0}$ through an initial coarse-graining pipeline $f_{cg} = f_{synth} \circ f_{seg}$. Each episodic unit is mapped to a structured state

$$d = (\lambda_{fact}, \Lambda_{conc}), \quad (4)$$

where $\lambda_{fact}$ denotes an objective event-level fact and $\Lambda_{conc}$ is a set of user-related conclusions. The conclusion set is partitioned as $\Lambda_{conc} = \Lambda_{base} \cup \Lambda_{rel}$, separating directly grounded interpretations from high-salience signals used for higher-level abstraction.

**Structured Knowledge Space $\mathcal{G}$.** A hierarchical extraction

function

$$f_{extract} : \mathcal{D}_{L0} \rightarrow \mathcal{G} \qquad (5)$$

organizes microscopic evidence into a dynamic knowledge graph $\mathcal{G} = (\mathcal{V}, \mathcal{E})$. The node set $\mathcal{V}$ is divided into three levels: abstract concepts $\mathcal{V}_{abs}$, representing high-level and slowly evolving user concepts; general events $\mathcal{V}_{gen}$, capturing recurring activities or patterns; and instance events $\mathcal{V}_{inst}$, corresponding to concrete, context-specific interactions. Edges $\mathcal{E}$ encode relationships between nodes and are partitioned into static classification relations $\mathcal{E}_{cls}$ and dynamic event relations $\mathcal{E}_{evt}$.

This separation provides a stable topological backbone for long-term abstractions while allowing event-driven relations to evolve incrementally. As a result, RGMem can propagate stable representations upward through the hierarchy while selectively updating event-level information in response to new evidence.

### 3.3.2. INSTANTIATION OF RG OPERATORS

The effective user profile in RGMem evolves through a set of explicit, scale-aware memory operators that incrementally integrate new evidence, refine abstractions, and propagate higher-level structure across the knowledge graph, enabling stable and interpretable profile formation under continuous and conflicting inputs.

**Relation Inference Operator** $\mathcal{R}_{K1}$**.** This operator performs incremental integration at the relation level, aggregating repeated or reinforcing evidence associated with the same semantic relation over time. Its primary role is to prevent uncontrolled accumulation of redundant facts while enabling stable, interpretable summaries of recurring user behaviors or states. Let $\mathcal{T}_e^{(1,t)}$ denote the mesoscopic theory associated with relation $e$ at time step $t$. When sufficient new evidence has accumulated for $e$, the operator updates the relation-level representation according to:

$$\mathcal{T}_e^{(1,t+1)} \leftarrow \mathcal{T}_e^{(1,t)} + \beta(\mathcal{T}_e^{(1,t)}, D_e^{new}), \qquad (6)$$

where $D_e^{new}$ is the set of newly observed microscopic evidence linked to relation $e$. The non-linear update function $\beta(\cdot)$, instantiated by a language model, integrates prior summaries with new evidence to produce an updated relation theory $\mathcal{T}_e^{(1,t+1)}$.

To avoid premature abstraction from sparse or noisy signals, $\mathcal{R}_{K1}$ is triggered only when the accumulated evidence for a relation exceeds a threshold $\theta_{\inf}$. This thresholded execution ensures that relation-level summaries emerge from consistent patterns rather than isolated observations, providing a stable intermediate representation for higher-level abstraction.

**Node-Level Abstraction Operator** $\mathcal{R}_{K2}$**.** While $\mathcal{R}_{K1}$ consolidates repeated evidence at the relation level, long-term

user profiling requires abstraction across multiple related behaviors and events. Operator $\mathcal{R}_{K2}$ performs this higher-order integration by aggregating information associated with an abstract concept node, producing a compact representation of user tendencies within a semantic domain.

For an abstract node $v \in \mathcal{V}_{abs}$, the operator consumes a mixed-scale input set

$$\mathcal{I}_v^{new} = \{\mathcal{T}_{e_i}^{(1),new}\}_{e_i \in N(v)} \cup \{d_j^{new}\}_{j \in D(v)}, \qquad (7)$$

The input includes recently updated relation-level summaries and newly observed microscopic evidence associated with $v$. Rather than treating all inputs equally, $\mathcal{R}_{K2}$ prioritizes aggregated representations to guide abstraction toward stable intermediate structures. Accordingly, $\mathcal{R}_{K2}$ performs an RG-inspired coarse-graining step that integrates heterogeneous evidence into a compact concept-level representation, and is decomposed into two sequential sub-operations:

$$\mathcal{R}_{K2} = \mathbb{S} \circ \mathbb{P}, \qquad (8)$$

corresponding to projection-selection and synthesis-rescaling, respectively.

**Projection–Selection** ($\mathbb{P}$)**.** Given the mixed-scale input set $\mathcal{I}_v^{new}$, the projection-selection step filters and prioritizes evidence according to its level of abstraction and information density. Relation-level theories that have already undergone aggregation are favored over raw microscopic observations, reflecting their higher semantic stability.

Formally, the filtered evidence set is defined as:

$$D_v' = \mathbb{P}(\mathcal{I}_v^{new}), \qquad (9)$$

where $D_v'$ contains a reduced subset of evidence that best represents collective behavioral signals associated with node $v$.

**Synthesis–Rescaling** ($\mathbb{S}$)**.** The synthesis step integrates the filtered evidence $D_v'$ with the previous node-level representation to construct an updated concept-level theory:

$$(\Sigma_v^{(2,t+1)}, \Delta_v^{(2,t+1)}) = \mathbb{S}(D_v', \Sigma_v^{(2,t)}, \Delta_v^{(2,t)}). \qquad (10)$$

**Order Parameter** $\Sigma$**.** The order parameter $\Sigma$ captures dominant, recurring patterns that persist across multiple situations. It represents the most stable abstraction of user behavior within a concept, prioritizing internal consistency and low representational complexity.

**Correction Term** $\Delta$**.** The correction term $\Delta$ preserves salient but non-universal signals that cannot be absorbed into the dominant abstraction without loss of fidelity. It explicitly represents internal tension within the profile, allowing the model to acknowledge conflicting or transitional behaviors.

These components are computed via non-linear aggregation functions:

$$\Sigma_v^{(2,t+1)} = \text{Agg}_{\text{common}}(D_v', \Sigma_v^{(2,t)}), \quad (11)$$

$$\Delta_v^{(2,t+1)} = \text{Extract}_{\text{salient}}(D_v', \Delta_v^{(2,t)}), \quad (12)$$

where both functions are instantiated by language models.

To prevent premature abstraction, $\mathcal{R}_{K2}$ is executed only when sufficient new evidence accumulates for a node, controlled by a threshold $\theta_{\text{sum}}$.

**Hierarchical Flow Operator $\mathcal{R}_{K3}$.** This operator captures the iterative nature of renormalization by propagating information upward along the static conceptual hierarchy $\mathcal{E}_{cls}$. It operates on macroscopic representations associated with abstract nodes, progressively integrating child-level summaries into higher-level profiles. At each level, the operator aggregates two complementary components $(\Sigma, \Delta)$.

Formally, for each parent node $v_p$, the transformation integrates representations from its child nodes $\{v_{c_i}\}$ as:

$$(\Sigma_{v_p}^{(s+1)}, \Delta_{v_p}^{(s+1)}) = \mathcal{R}_{K3}(\{(\Sigma_{v_{c_i}}^{(s)}, \Delta_{v_{c_i}}^{(s)})\}_i), \quad (13)$$

where $\mathcal{R}_{K3}$ performs a structured synergy–tension analysis to distill common patterns while retaining critical residual signals. The execution of this operator is scheduled via a dirty-flag propagation mechanism to ensure efficient and incremental updates.

### 3.4. Dynamics and Multi-Scale Observations

**Emergent Dynamics: Stability and Structural Shifts** When applied continually over long interaction streams, RGMem exhibits two characteristic dynamical behaviors that govern the evolution of user profiles.

First, under consistent and reinforcing evidence, higher-level profile representations gradually stabilize. Updates induced by new interactions diminish over time, leading to a regime where macroscopic profile states remain effectively invariant. In practice, this corresponds to a stable long-term user profile that is robust to routine or redundant inputs.

Second, when accumulated evidence becomes sufficiently strong and inconsistent with the current profile, the system undergoes a qualitative reorganization. Instead of incremental adjustment, higher-level representations are restructured in a coordinated manner, resulting in a rapid shift of the user profile. These structural shifts are triggered by thresholded update mechanisms and reflect genuine changes in user preferences rather than transient noise.

Together, these two regimes enable RGMem to balance long-term stability with responsiveness to change. As we show empirically in Section 4, this behavior manifests as non-monotonic performance under varying update thresh-

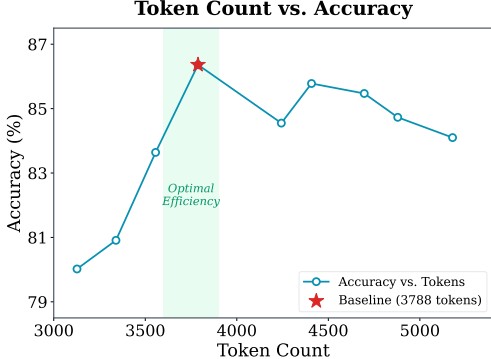

*Figure 3.* Accuracy peaks at an effective context length

olds and allows RGMem to simultaneously achieve strong stability and plasticity.

**Multi-Scale Observations** RGMem exposes its internal memory state through a multi-scale retrieval mechanism implemented in the L2 layer. Given a query $q$, the retrieval function $f_{\text{retr}}(q, \mathcal{M})$ constructs a query-specific context by selectively accessing memory representations at different abstraction levels.

At the microscopic level, the retriever returns relevant episodic evidence to support fact-level queries. At higher levels, it retrieves aggregated relational or profile-level representations that summarize longer-term patterns. These components are combined into a unified context $C(q)$ and provided to the language model for response generation.

This design allows RGMem to adapt the granularity of retrieved information to the query intent, enabling both precise factual recall and abstract reasoning over long-term user traits within a bounded context.

## 4. Experiment Evaluation

### 4.1. Experimental Setup

**Benchmarks and Baselines.** We evaluate RGMem on two long-term conversational memory benchmarks: LOCOMO (Maharana et al., 2024) and PersonaMem under the 128k-token setting (Jiang et al., 2025). LOCOMO focuses on long-context reasoning and temporal consistency, while PersonaMem evaluates dynamic persona evolution under conflicting evidence. Detailed benchmark descriptions, evaluation protocols, and implementation details are provided in Appendix B.1 and B.10.

### 4.2. Memory Efficiency and Information Density

We examine whether hierarchical coarse-graining improves the effective information density of long-term conversational memory under limited context budgets. Using the

*Table 1.* Performance on the PersonaMem benchmark (%). We report the improvement of our method compared to the second-best baseline in parentheses.

| Backbone | Method | MEMORY RECALL | | | REASONING & ADAPTATION | | | TEMPORAL | |
| | | Recall Facts | Suggest Ideas | Revisit Reasons | Track Evol. | Aligned Rec. | General. Scen. | Latest Pref. | Avg. |
|---|---|---|---|---|---|---|---|---|---|
| *Backbone: GPT-4o-mini* | | | | | | | | | |
| | LLM (Vanilla) | 55.34 | 12.33 | 70.02 | 58.21 | 42.33 | 33.05 | 34.31 | 39.21 |
| | LangMem | 64.76 | 23.93 | 81.82 | 53.30 | 45.45 | 42.26 | 58.12 | 52.36 |
| | Mem0 | 69.57 | 19.53 | 78.75 | 56.72 | 61.82 | 52.19 | 68.25 | 56.79 |
| | A-Mem | 59.78 | 9.22 | 80.19 | 53.30 | 44.32 | 45.42 | 53.25 | 49.17 |
| | Memory OS | 72.59 | 22.62 | 84.42 | **69.11** | 67.82 | 35.72 | 64.71 | 54.23 |
| | **RGMem** | **77.06** (+4.47) | **26.47** (+2.54) | **85.29** (+0.87) | 67.82 | **73.66** (+5.84) | **56.62** (+4.43) | **75.47** (+7.22) | **63.87** (+7.08) |
| *Backbone: GPT-4.1* | | | | | | | | | |
| | LLM (Vanilla) | 64.76 | 19.53 | 81.82 | 67.21 | 53.14 | 53.91 | 51.62 | 51.86 |
| | LangMem | 77.82 | 24.27 | 82.16 | 54.33 | 42.33 | 63.22 | 70.59 | 58.23 |
| | Mem0 | 81.02 | 16.28 | 81.82 | 54.33 | 54.35 | 65.13 | 81.22 | 60.44 |
| | A-Mem | 82.57 | 28.31 | 86.35 | 56.72 | 70.12 | 65.13 | 77.81 | 63.95 |
| | Memory OS | 79.72 | 19.33 | 82.16 | 60.34 | 73.66 | **74.18** | 77.81 | 65.03 |
| | **RGMem** | **88.64** (+6.07) | **35.04** (+6.73) | **87.03** (+0.68) | **70.89** (+3.68) | **83.02** (+9.36) | 72.55 | **85.07** (+3.85) | **74.01** (+8.98) |

*Table 2.* LOCOMO benchmark results (%). LLM-as-a-judge evaluation.

| Method | QUESTION TYPE | | | | |
| | S-Hop | M-Hop | Open | Temp. | Avg. |
|---|---|---|---|---|---|
| *gpt-4o-mini* | | | | | |
| RAG | 35.05 | 30.31 | 43.52 | 27.58 | 38.10 |
| LangMem | 62.23 | 47.92 | 71.12 | 23.43 | 58.10 |
| Mem0 | 67.13 | 51.15 | **72.93** | 55.51 | 66.88 |
| Zep | 74.11 | 66.04 | 67.71 | 79.76 | 75.14 |
| **RGMem** | **80.15** | **69.16** | 67.71 | **81.27** | **78.92** |
| *Full-Context* | *83.01* | *66.79* | *49.53* | *58.22* | *71.41* |
| *gpt-4.1-mini* | | | | | |
| RAG | 37.94 | 37.69 | 48.96 | 61.83 | 51.62 |
| LangMem | 74.47 | 61.06 | 67.71 | 86.92 | 78.05 |
| Mem0 | 62.41 | 57.32 | 44.79 | 66.47 | 62.47 |
| Zep | 79.43 | 69.16 | **73.96** | 83.33 | 79.09 |
| **RGMem** | **89.58** | **78.03** | 72.86 | **88.91** | **86.17** |
| *Full-Context* | *88.53* | *77.70* | *71.88* | *92.70* | *87.52* |

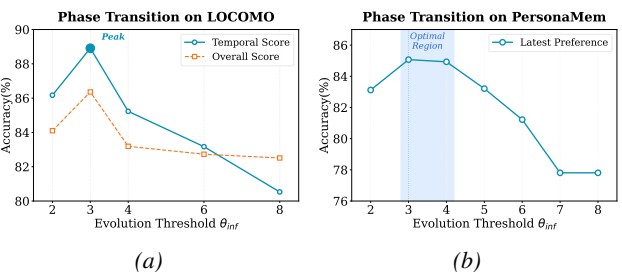

*Figure 4.* Non-linear dependence on evolution threshold.

LOCOMO benchmark, we analyze how reasoning performance varies with the amount of retrieved context. Fig. 3 shows a clear non-monotonic relationship between context length and accuracy. Performance improves as context increases from approximately 3k to 3.8k tokens, but saturates and then degrades as more context is added. This indicates that indiscriminate accumulation of historical information does not yield monotonic gains in reasoning quality. Instead, the results reveal an optimal context scale at which information is maximally useful. Below this scale, evidence is insufficient; beyond it, redundant or weakly relevant content obscures salient signals. RGMem operates near this regime by hierarchically coarse-graining dialogue history into compact, high-density representations rather than relying on flat retrieval or full-context accumulation. From a renormalization perspective, this behavior reflects systematic scale selection, where irrelevant microscopic fluctuations are integrated out through coarse-graining and rescaling, yielding an effective description that preserves task-relevant invariants. These results support **Insight 1**: effective information density is maximized through hierarchical abstraction, not brute-force context expansion.

### 4.3. Evidence of Phase-Transition Dynamics

Standard memory systems typically exhibit monotonic behavior, where updates lead to either gradual improvement or instability. In contrast, the RG perspective predicts nonlinear dynamics governed by control parameters. We an-

alyze RGMem's behavior as a function of the evolution threshold $\theta_{\text{inf}}$, which acts as a **control parameter**, while task-level performance serves as an observable **order parameter** reflecting the macroscopic state of the user profile.

Fig. 4 reveals a sharp, non-monotonic performance peak at $\theta_{\text{inf}} = 3$ across both LOCOMO (Fig. 4a) and PersonaMem (Fig. 4b). Rather than a smooth optimum, this behavior indicates the presence of a *critical point* in the memory dynamics.

**Two Regimes and a Critical Point.** The system exhibits two qualitatively distinct regimes: **Subcritical Regime ($\theta_{\text{inf}} < 3$):** The system is overly sensitive, where even transient noise triggers frequent updates, leading to **Supercritical Regime ($\theta_{\text{inf}} > 3$):** Updates are excessively suppressed, causing the profile to become rigid and unresponsive to genuine preference changes.

At the critical threshold ($\theta_{\text{inf}} = 3$), RGMem achieves an optimal balance. Below this point, new observations are absorbed as minor perturbations; once accumulated evidence crosses the threshold, the system undergoes a rapid and coordinated reorganization of its macroscopic state. This phase-transition-like behavior enables robustness to noise while remaining responsive to meaningful shifts.

**Universality and Implications. Universality and Implications.** Despite differences in tasks and metrics, both benchmarks exhibit the same critical threshold, indicating that this behavior reflects an intrinsic property of multi-scale memory dynamics rather than dataset-specific artifacts. Operating near this critical point enables RGMem to suppress fast-varying noise while preserving slow-varying, task-relevant structure, providing a principled resolution to the stability–plasticity dilemma. From a renormalization perspective, this corresponds to integrating out irrelevant fluctuations while retaining the macroscopic components that define long-term user state. These results support **Insight 2**: user profile evolution follows a non-linear, threshold-driven dynamic.

### 4.4. Stability–Plasticity Trade-off and Emergent Macroscopic Invariance

Overall accuracy alone does not reveal how memory systems balance long-term consistency with rapid adaptation. To explicitly examine this tension, we analyze the stability–plasticity trade-off by jointly visualizing performance on Recall Facts and Latest Preference in Fig. 5.

As shown in the figure, most baseline methods lie on a clear Pareto frontier. Systems that aggressively update memory improve adaptability but sacrifice factual retention, while more conservative designs preserve historical information at the cost of responsiveness. In contrast, **RGMem** lies strictly beyond this frontier, achieving superior performance on both dimensions simultaneously. This indicates that RGMem does not merely shift the trade-off curve, but effectively breaks the conventional stability–plasticity constraint.

This behavior arises from RGMem's separation of memory across scales. Factual evidence is retained at lower levels to ensure stability, while higher-level abstractions evolve selectively based on aggregated evidence, enabling plasticity without overreacting to noise. By decoupling slow-varying traits from fast-changing observations, RGMem avoids forcing information to evolve at a single scale.

More broadly, these results suggest that robustness in long-term user modeling stems from the emergence of *macroscopic invariants*, rather than fact-level stability. Tasks involving multi-hop reasoning or cross-scenario generalization benefit from abstract behavioral regularities that remain stable across contexts. In RGMem, such regularities function as order parameters,while detailed facts, acting as correction terms, preserve fdelity without dominating inference. These fndings well-support **Insight 3**: separating slow and fast variables resolves the stability–plasticity dilemma and **Insight 4**: long-term iser profiles exhibit macroscopic invariance beyond fact-level stability.

### 4.5. Additional Analyses

Appendix experiments include ablations (Appendix B.4), parameter sensitivity analysis (Appendix B.8), and an analysis of macroscopic profile evolution (Appendix B.6). Ablation results show that removing any core component of the multi-scale memory design consistently degrades performance, even when more context is retrieved. Parameter sensitivity analysis indicates that RGMem remains stable across a broad range of retrieval budgets and evolution thresholds, with a clear optimal regime. Finally, the macroscopic profile evolution analysis reveals that under consistent long-term evidence, profile representations rapidly converge and stabilize, exhibiting attractor-like behavior.

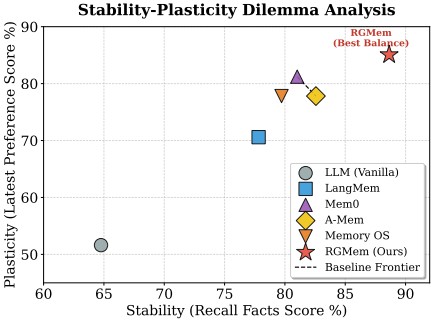

*Figure 5.* Stability–plasticity trade-off on PersonaMem.

## 5. Conclusion

We have studies long-term conversational memory as a multi-scale dynamical system rather than a static retrieval problem. Our multi-facet experiments on the representative datasets LOCOMO and PersonaMem reveal that effective long-term user-profile modeling depends on hierarchical coarse-graining, thresholded non-linear updates, separation of slow and fast variables, and macroscopic invariance beyond fact-level stability. Motivated by these fundamental findings, we propose a novel self-evolving memory framework RGMem, enabling to instantiate scale-aware abstraction and control evolution through explicit memory structures. This work unveils that a robust long-term personalization in language agents requires principled multi-scale organization, rather than larger context windows or fat memory accumulation.

## Impact Statement

This paper presents work whose goal is to advance the field of Machine Learning. There are many potential societal consequences of our work, none which we feel must be specifically highlighted here.

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

# Appendices

Within this supplementary material, we elaborate on the following aspects:

- Appendix A: Algorithm

- Appendix B: Additional Experimental Results

    - Appendix B.1: Benchmarks and Evaluation Protocol

    - Appendix B.2: Full Results on the LOCOMO Benchmark

    - Appendix B.3: Robustness on Dynamic Persona Evolution (PersonaMem)

    - Appendix B.4: Ablation Study: Validating Multi-Scale RG-Inspired Dynamics

    - Appendix B.5: Comparison of Profile Update Strategies

    - Appendix B.6: Temporal Convergence and Fixed-Point Analysis of Macroscopic User Profiles

    - Appendix B.7: Robustness to Abrupt Profile Mutations and Preference Shifts

    - Appendix B.8: Parameter Sensitivity Analysis

    - Appendix B.9: Computational Cost and Efficiency Analysis

    - Appendix B.10: Implementation Details

- Appendix C: Qualitative Case Study of Error Isolation in Memory Evolution

- Appendix D: Theoretical Analysis

    - Proposition D.1: Coarse-grained majority summary increases information

    - Proposition D.2: Thresholded profile dynamics induce phase-transition–like updates

    - Proposition D.3: Two-Timescale Update: Stability and Plasticity

- Appendix E: Prompt Templates

# A. Algorithm

---

**Algorithm 1** RGMem: Continual Memory Evolution and Multi-Scale Retrieval

---

1: **Input:** dialogue stream $D_{\mathrm{raw}} = \{u_1, \ldots, u_T\}$, query set $Q$, thresholds $(\theta_{inf}, \theta_{sum})$
2: **Output:** final memory state $\mathcal{M} = (\mathcal{D}_{L0}, \mathcal{G})$, answers $\{a_q\}_{q \in Q}$
3: Initialize $\mathcal{D}_{L0} \leftarrow \emptyset$
4: Initialize $\mathcal{G} \leftarrow (\mathcal{V}_{abs}, \mathcal{V}_{gen}, \mathcal{V}_{inst}, \mathcal{E}_{cls}, \mathcal{E}_{evt})$ with base abstract nodes
5: **// Continual RG evolution along the dialogue**
6: **for** $t = 1$ **to** $T$ **do**
7:     $S_t \leftarrow f_{seg}(u_t)$
8:     **for all** $s \in S_t$ **do**
9:         $d \leftarrow f_{synth}(s)$                                        // $d = (\lambda_{fact}, \Lambda_{conc})$
10:        $\mathcal{D}_{L0} \leftarrow \mathcal{D}_{L0} \cup \{d\}$
11:        paths $\leftarrow$ EXTRACT_ENTITIES($d$)
12:        triples $\leftarrow$ EXTRACT_RELATIONS($d$, paths)
13:        $(\mathcal{V}_{abs}, \mathcal{V}_{gen}, \mathcal{V}_{inst}, \mathcal{E}_{cls}, \mathcal{E}_{evt}) \leftarrow$ UPDATE_GRAPH($\mathcal{V}_{abs}, \mathcal{V}_{gen}, \mathcal{V}_{inst}, \mathcal{E}_{cls}, \mathcal{E}_{evt}$, paths, triples)
14:        update local buffers $D_e^{new}$, $\mathcal{I}_v^{new}$, and counters mentions($e$), score($v$)
15:     **end for**
16:     **// Symmetric scheduling of RG operators on new evidence**
17:     **for all** $e \in \mathcal{E}_{evt}$ **with** mentions($e$) $\geq \theta_{inf}$ **do**
18:        $\mathcal{T}_e \leftarrow \mathcal{R}_{K1}(\mathcal{T}_e, D_e^{new})$
19:        $D_e^{new} \leftarrow \emptyset$, mentions($e$) $\leftarrow 0$
20:     **end for**
21:     **for all** $v \in \mathcal{V}_{abs}$ **with** score($v$) $\geq \theta_{sum}$ **do**
22:        $(\Sigma_v, \Delta_v) \leftarrow \mathcal{R}_{K2}((\Sigma_v, \Delta_v), \mathcal{I}_v^{new})$
23:        $\mathcal{I}_v^{new} \leftarrow \emptyset$, score($v$) $\leftarrow 0$
24:        mark parent($v$) as dirty
25:     **end for**
26:     **for all** dirty $v_p$ in topological order **do**
27:        $(\Sigma_{v_p}, \Delta_{v_p}) \leftarrow \mathcal{R}_{K3}(\{(\Sigma_{v_c}, \Delta_{v_c})\}_{v_c \in \mathrm{child}(v_p)})$
28:        mark $v_p$ as clean
29:     **end for**
30: **end for**
31: **// Multi-scale retrieval for queries**
32: **for all** $q \in Q$ **do**
33:    entity_queries $\leftarrow$ QUERY_STRUCTURING($q$)
34:    $L0\_ctx \leftarrow$ BM25_RETRIEVE($\mathcal{D}_{L0}$, entity_queries)
35:    $L1\_ctx \leftarrow$ KG_RETRIEVE($\mathcal{G}$, entity_queries)
36:    $C(q) \leftarrow$ FORMAT($L0\_ctx$, $L1\_ctx$)
37:    $a_q \leftarrow$ LLM($q$, $C(q)$)
38: **end for**
39: **return** $\mathcal{M} = (\mathcal{D}_{L0}, \mathcal{G})$, $\{a_q\}_{q \in Q}$

---

# B. Additional Experimental Results

## B.1. Benchmarks and Evaluation Protocol

**Evaluation Protocol Clarification.**    We report benchmark-specific evaluation metrics following prior work. On LOCOMO, performance is measured using an LLM-as-a-judge protocol, where answers are evaluated by `gpt-4.1` as the judging model. On PersonaMem, all questions are multiple-choice, and performance is reported as accuracy.

All experiments are run three times with different random seeds, and the reported results are averaged across runs. On LOCOMO, we evaluate RGMem and baselines using `gpt-4o-mini` and `gpt-4.1-mini` as backbone models. On PersonaMem, we evaluate using `gpt-4o-mini` and `gpt-4.1` backbones.

**LOCOMO.** We conduct experiments on the Long-term Conversational Memory (LOCOMO) benchmark (Maharana et al., 2024), which is designed to evaluate memory and reasoning over ultra-long conversation histories. The dataset consists of 10 independent conversations, each containing approximately 600 interaction turns and 26k tokens, along with around 200 question–answer pairs. The evaluation protocol provides the complete conversation history to the system for memory construction, followed by a sequence of queries. Questions are categorized into Single-Hop, Multi-Hop, Temporal, and Open-Domain reasoning, enabling fine-grained analysis of different long-context reasoning behaviors. Performance is measured using an LLM-as-a-judge protocol consistent with prior work.

**PersonaMem.** PersonaMem (Jiang et al., 2025) is a dialogue-based benchmark designed to evaluate long-term memory under dynamically evolving user personas. User preferences, habits, and personal facts may change over time, often with explicit conflicts between earlier and later information. We adopt the most challenging 128k-token configuration, where the full interaction history can span up to 128,000 tokens. PersonaMem categorizes evaluation questions into seven fine-grained skill types:

- **Recall User-Shared Facts**: assessing the ability to retrieve static personal information mentioned earlier.

- **Acknowledge Latest User Preferences**: evaluating whether the model correctly follows the most recent user state when conflicts arise.

- **Track Full Preference Evolution**: testing the ability to summarize how preferences change over time.

- **Revisit Reasons Behind Preference Updates**: requiring causal reasoning over events that triggered state changes.

- **Provide Preference-Aligned Recommendations**: assessing whether recommendations align with the current user profile.

- **Suggest New Ideas**: evaluating the ability to propose novel but relevant options not previously mentioned.

- **Generalize to New Scenarios**: testing cross-domain transfer of inferred user traits.

We report accuracy for each dimension as well as the Overall score.

**Baselines.** To ensure fair comparison, we follow established experimental settings from prior work and exclude adversarial or unanswerable query categories. Across both benchmarks, we evaluate against representative explicit memory systems, including LangMem and Mem0 (Chhikara et al., 2025). For LOCOMO, we additionally include standard retrieval-augmented generation (RAG-500), Zep (Rasmussen et al., 2025), and a Full-Context setting that provides the entire dialogue history as a theoretical upper bound. For PersonaMem, we compare against a vanilla LLM using context window only, as well as recent agentic memory frameworks such as A-Mem-(Xu et al., 2026) and MemoryOS-(Li et al., 2025). All baseline implementations follow their original descriptions, and evaluation metrics are kept consistent across methods.

### B.2. Full Results on the LOCOMO Benchmark

This section reports the complete experimental results on the LOCOMO benchmark, which are omitted from the main text due to space constraints. LOCOMO evaluates long-term conversational memory over ultra-long dialogue histories and covers four complementary reasoning categories: Single-Hop, Multi-Hop, Temporal, and Open-Domain reasoning.

Table 2 presents the full comparison between RGMem and representative baseline memory systems under different language model backbones. As shown in the table, RGMem consistently achieves the strongest overall performance and significantly outperforms retrieval-based and flat memory approaches.

**Overall Performance.** Using `gpt-4.1-mini` as the backbone, RGMem achieves an overall accuracy of 86.17%, exceeding the strongest baseline (Zep, 79.09%) by over 7 percentage points and closely approaching the theoretical upper bound of the Full-Context setting (87.52%). This demonstrates that RGMem is able to match near-oracle performance without relying on full-context accumulation.

**Single-Hop Reasoning.** RGMem attains a Single-Hop accuracy of 89.58%, slightly surpassing the Full-Context upper bound. This result highlights the benefit of the initial coarse-graining stage, which filters noisy raw dialogue into high signal-to-noise factual evidence and user conclusions. In contrast, providing the entire dialogue history introduces redundancy and distractors that hinder precise fact localization.

**Multi-Hop Reasoning.** The largest performance margin is observed in Multi-Hop reasoning, where RGMem outperforms the best baseline by nearly 9 percentage points. This improvement reflects RGMem's ability to consolidate dispersed evidence through relational abstraction and hierarchical evolution, reducing the integration burden at inference time.

**Temporal Reasoning.** RGMem also achieves the highest Temporal reasoning score (88.91%), validating the effectiveness of its thresholded evolution mechanism in tracking preference changes and resolving temporal conflicts.

**Open-Domain Reasoning.** Performance gains on Open-Domain queries are comparatively smaller. This is consistent with RGMem's design objective of prioritizing stable, task-relevant user traits over high-entropy or weakly related information. Despite this, RGMem remains competitive with strong baselines in this category.

Overall, these results corroborate the main text findings by showing that RGMem's advantages stem from structured multi-scale organization and controlled memory evolution, rather than brute-force context accumulation.

### B.3. Robustness on Dynamic Persona Evolution (PersonaMem)

This section reports the full experimental results on the PersonaMem benchmark, which evaluates long-term personalized memory under dynamically evolving user profiles. PersonaMem is designed to stress-test a system's ability to track preference shifts, resolve conflicts between outdated and recent information, and generalize across scenarios over up to 60 dialogue sessions.

Table 2 summarizes the performance of RGMem and competing memory systems across seven evaluation dimensions.

**Overall Performance.** With the `gpt-4.1` backbone, RGMem achieves an overall accuracy of 74.01%, substantially outperforming strong baselines such as Memory OS (65.03%) and A-Mem (63.95%). Compared to the vanilla LLM (51.86%), RGMem improves performance by over 22 percentage points, demonstrating the necessity of explicit long-term memory organization under dynamic conditions.

**Tracking Preference Evolution.** RGMem shows particularly strong performance on *Acknowledge Latest User Preference* and *Track Full Preference Evolution*. These tasks require distinguishing genuine preference shifts from transient noise, a scenario where many baseline systems struggle due to recency bias or unstructured accumulation. RGMem's thresholded evolution mechanism enables it to update user profiles decisively only when sufficient evidence accumulates.

**Generalization and Causal Reasoning.** RGMem also achieves high accuracy on *Generalize to New Scenarios* and *Revisit Reasons behind Updates*. These results indicate that RGMem captures abstract behavioral regularities that transfer across contexts, while preserving salient causal evidence that explains why preferences changed.

**Summary.** The PersonaMem results provide complementary evidence to the LOCOMO benchmark. While LOCOMO emphasizes long-horizon reasoning over a fixed conversation, PersonaMem explicitly evaluates dynamic evolution under conflicting evidence. Across both settings, RGMem consistently demonstrates robust adaptation, long-term consistency, and superior generalization, supporting the central claim that long-term user memory should be modeled as a multi-scale, thresholded dynamical process.

### B.4. Ablation Study: Validating Multi-Scale RG-Inspired Dynamics

This section analyzes the contribution of core components in RGMem. Rather than viewing these ablations as isolated architectural choices, we interpret them as empirical evidence for the necessity of **RG-inspired design constraints**—specifically, *scale separation* and *non-linear, thresholded evolution*—in inducing robust long-term memory dynamics. We emphasize that RG is adopted here as an *engineering principle*, rather than as a physical theory, to impose structured constraints on how memory representations evolve across abstraction levels.

*Table 3.* We evaluate ablated variants that remove key components of RGMem. Removing the L0 layer degrades performance across all categories, while removing the L1 layer primarily affects multi-hop and temporal reasoning. The `w/o RG` variant replaces RG-inspired scale-dependent evolution with single-scale updates, leading to lower accuracy despite substantially higher token usage. RGMem achieves the best performance with more efficient context utilization. Accuracy is reported in percentage (%).

| | QUESTION TYPE | | | | SUMMARY | |
|---|---|---|---|---|---|---|
| Method | S-Hop | M-Hop | Open | Temp | Overall | Avg.Tok. |
| *gpt-4.1-mini* | | | | | | |
| w/o L0 | 58.58 | 41.48 | 52.51 | 65.26 | 56.29 | **1,885** |
| w/o L1 | 85.90 | 65.59 | 64.60 | 81.87 | 79.88 | 2,068 |
| w/o RG | 87.21 | 69.23 | 63.05 | 84.22 | 82.17 | 4,354 |
| **RGMem** | **89.58** | **78.03** | **72.86** | **88.91** | **86.17** | 3,788 |

*Table 4.* The `w/o L0`, `w/o L1`, and `w/o RG` variants remove the factual layer, relational abstraction, and RG-inspired evolution, respectively. **Overall** reports the average performance across evaluation dimensions, and **Avg. Tokens** denotes the average retrieved context length. The `w/o RG` variant underperforms RGMem despite using substantially more tokens, indicating the importance of thresholded, multi-scale evolution over uniform updates or increased context budgets.

| | MEMORY RECALL | | | REASONING & ADAPTATION | | | TEMPORAL | SUMMARY | |
|---|---|---|---|---|---|---|---|---|---|
| Method | Recall Facts | Suggest Ideas | Revisit Reasons | Track Evol. | Aligned Rec. | General. Scen. | Latest Pref. | Overall | Avg. Tokens |
| *Backbone: GPT-4o-mini* | | | | | | | | | |
| w/o L0 | 34.35 | **28.31** | 72.23 | 47.82 | 70.12 | 43.79 | 62.31 | 47.22 | **2,355** |
| w/o L1 | 70.22 | 15.26 | 77.79 | 50.17 | 59.85 | 41.22 | 60.75 | 54.31 | 4,267 |
| w/o RG | 72.35 | 24.27 | 81.82 | 56.72 | 63.54 | 47.15 | 67.88 | 57.37 | 8,479 |
| **RGMem** | **77.06** | 26.47 | **85.29** | **67.82** | **73.66** | **56.62** | **75.47** | **63.87** | 7,105 |

### B.4.1. THE NECESSITY OF HIERARCHICAL ABSTRACTION AND SCALE SEPARATION

**Ablation on LOCOMO: The Role of Scale Separation.** Table 3 reports ablation results on the LOCOMO benchmark.

- **Loss of Fine-Grained Evidence (w/o L0):** Removing the factual evidence layer leads to severe degradation across all reasoning categories. This confirms that higher-level profile representations must be grounded in stable fine-grained evidence. Without such a microscopic basis, abstract summaries become unreliable and prone to hallucination.

- **Breakdown of Hierarchical Coarse-Graining (w/o L1):** Removing the relational abstraction layer primarily impairs Multi-Hop and Temporal reasoning. This indicates that directly mapping isolated facts to high-level traits is insufficient. The L1 layer functions as a necessary coarse-graining step, constructing meso-scale relational structures that mediate between transient observations and more stable profile representations.

- **Failure of Single-Scale Summarization (w/o RG):** The `w/o RG` variant replaces structured, scale-dependent evolution with flat prompting and uniform updates. While this variant partially recovers performance by substantially increasing retrieved context length, it still consistently underperforms the full RGMem and incurs significantly higher token consumption. This result demonstrates that brute-force context accumulation cannot substitute for principled multi-scale organization. Without explicit rescaling and filtering of weakly relevant information, the effective higher-level representation becomes cluttered with noise, reducing information density and reasoning reliability.

**Ablation on PersonaMem: Dynamics and Stability–Plasticity.** Table 4 presents complementary results on the Person-aMem benchmark, which explicitly stresses long-term persona evolution under conflicting and outdated evidence.

- **Decoupling Slow- and Fast-Varying Representations:** The full RGMem substantially outperforms the `w/o RG` variant on *Track Evolution* and *Generalize Scenario*. This gap highlights the importance of separating slowly evolving, high-level user traits from rapidly changing factual observations. Without such separation, memory updates either overreact to transient noise or fail to adapt to genuine preference shifts.

- **Absence of Thresholded Regime Changes:** The `w/o RG` variant exhibits consistent degradation across dynamic dimensions. Lacking thresholded update mechanisms (e.g., $\theta_{\text{inf}}$), the system fails to exhibit the non-linear, regime-change-like behavior observed in RGMem (Fig. 6). Instead, updates accumulate in a linear and uncoordinated manner, leading to unstable adaptation or excessive rigidity over long horizons.

**Conclusion.** Across both benchmarks, these ablations demonstrate that RGMem's performance gains do not arise from architectural redundancy or increased context budgets. Rather, they stem from the principled enforcement of **RG-inspired multi-scale dynamics**, including hierarchical coarse-graining, separation of slow and fast representations, and thresholded non-linear updates. When these constraints are removed, the system collapses into a single-scale incremental memory regime that cannot recover the same stability–plasticity balance, even with substantially larger context usage.

### B.4.2. ONTOLOGICAL STABILITY: FIXED POINTS VS. UNCONSTRAINED CLUSTERING

To further dissect the specific contributions of RGMem's architectural constraints, we conducted additional targeted ablations on a 20% subset of the PersonaMem benchmark. These experiments specifically evaluate the necessity of fixed semantic coordinates versus unconstrained clustering, as well as the critical role of separating the order parameter ($\Sigma$) from the correction term ($\Delta$).

Many hierarchical memory systems rely on dynamic graph construction and geometric clustering (e.g., KNN, semantic similarity) to form macroscopic abstractions from microscopic events. However, clustering typically captures only surface-level lexical or semantic similarity, often missing deeper causal or logical relations. Over very long dialogues, this lack of structural priors can lead to chaotic macro-nodes and semantic drift.

To address this, RGMem's design strictly separates fast variables (micro-details) from slow variables (macro-traits). Rather than employing unconstrained clustering, RGMem predefines abstract semantic nodes as "Fixed Points" within the knowledge graph. Micro-events are systematically mapped to these predefined dimensions.

*Table 5.* Performance comparison on a 20% subset of PersonaMem. We specifically report the metrics most sensitive to temporal macro-structures. L0+GraphRAG represents a baseline that applies unconstrained geometric clustering and summarization over RGMem's extracted L0 micro-facts.

| Method | Recall Facts | Track Pref. Evol. | Latest Pref. |
|---|---|---|---|
| L0+GraphRAG | 79.86 | 51.49 | 49.72 |
| **RGMem** | **81.65** | **68.77** | **74.11** |

To validate this design, we compared RGMem against a "L0+GraphRAG" baseline, which takes RGMem's L0 outputs (micro-facts) but utilizes standard GraphRAG-style clustering and summarization for higher levels. As shown in Table 5, while L0+GraphRAG retains basic facts adequately (79.86% Recall), its accuracy drops severely (by ∼17-25%) on dynamic tasks such as tracking evolution and acknowledging the latest preference. Relying solely on similarity-based clustering lacks stable semantic anchors, leading to progressive semantic drift over extended interactions. By anchoring micro-events to predefined conceptual dimensions, RGMem establishes a stable semantic coordinate system, significantly improving its ability to accurately track profile shifts.

### B.4.3. PREVENTING OVER-SMOOTHING: THE ROLE OF THE $(\Sigma, \Delta)$ TUPLE

Existing memory architectures frequently rely on recursive LLM summarization to compress long context. However, LLMs naturally favor "majority voting" during summarization, which often leads to information over-smoothing. During sudden user preference reversals, recent conflicting signals are easily "averaged away" by the sheer volume of historical conversation, causing the agent to exhibit sluggish adaptation or selective amnesia.

RGMem's coarse-graining module actively prevents this by preserving critical-point fluctuations. Instead of generating a single, flat summary, the operator explicitly extracts a tuple $(\Sigma, \Delta)$. This isolates new, conflicting shifts (the Tension/Correction term, $\Delta$) from the dominant historical traits (the Synergy/Order Parameter, $\Sigma$).

**Case Example:** Consider a memory state where the history contains dozens of strength training logs, but the latest turn indicates a shift to mild rehab exercises due to a back injury. A traditional recursive summarizer treats the recent injury as noise against the overwhelming fitness history, yielding a false "heavy fitness" profile. In contrast, RGMem extracts $\Sigma$ as

"consistent fitness habit" AND explicitly extracts $\Delta$ as "current physical constraint: back injury rehab", thereby fixing the over-smoothing issue and allowing downstream reasoning to respect the latest constraint.

*Table 6.* Ablation on the Tension/Correction term ($\Delta$) extraction, evaluated on a 20% subset of PersonaMem. The evaluation focuses exclusively on the *Latest Preference* metric, which is specifically designed to stress-test conflict resolution capabilities.

| Aggregation Mechanism | Latest Pref. (%) |
|---|---|
| w/o $\Delta$ extraction (Single Summary) | 52.58 |
| **with $(\Sigma, \Delta)$ extraction** | **74.11** |

As demonstrated in Table 6, removing the $\Delta$ extraction mechanism (forcing the model to output a single summary) causes the system to fail catastrophically when resolving conflicting states, dropping the Latest Preference accuracy to 52.58%. The $(\Sigma, \Delta)$ extraction fundamentally resolves this by explicitly maintaining internal representational tension, preventing irreversible information loss during compression.

## B.5. Comparison of Profile Update Strategies

A critical challenge in managing macroscopic user profiles is determining the optimal trigger for memory updates to avoid noise integration and delayed responses. Existing memory systems typically rely on rigid, external clocks. For instance, high-frequency incremental updates (e.g., updating per conversational turn) often inject transient noise, leading to representational oscillation and redundant computational overhead. Conversely, periodic global updates (e.g., scheduled daily or session-end rewrites) lag behind genuine preference shifts and are susceptible to pollution from localized, irrelevant chitchat.

Motivated by the critical point behavior in Renormalization Group theory, RGMem employs a local evidence density trigger. Updates in RGMem are event-driven and executed only when accumulated microscopic facts breach a predefined threshold (e.g., $\theta_{\text{inf}} = 3$). This non-linear mechanism allocates computational resources strictly to regions of the memory graph that require structural changes.

*Table 7.* Cost and accuracy comparison of different profile update mechanisms evaluated on a 20% subset of the PersonaMem benchmark.

| Update Mechanism | Input Tok. (k) | Output Tok. (k) | Acc. (%) |
|---|---|---|---|
| High-Frequency ($\theta_{\text{inf}} = 1$) | 384.23 | 63.09 | 53.72 |
| Periodic (Global Rewrite) | 237.76 | 39.87 | 48.95 |
| **RGMem (Threshold-Driven)** | **295.17** | **52.62** | **65.32** |

To empirically validate this design, we compared RGMem against two baseline update strategies on a 20% subset of the PersonaMem benchmark: (1) *High-Frequency* (updating upon every new observation, equivalent to $\theta_{\text{inf}} = 1$) and (2) *Periodic* (scheduled global rewrites at fixed turn intervals). As detailed in Table 7, RGMem's threshold-based trigger achieves significantly higher accuracy (65.32%) while maintaining a balanced computational footprint compared to the excessive token usage of high-frequency updates.

Furthermore, we tracked the representational stability of the macroscopic profiles by measuring the cosine similarity $\cos(\Sigma_t, \Sigma_{t-1})$ between consecutive updates over 763 dialogue turns (Fig. 8). The varying dynamics are evident:

- **High-Frequency:** Exhibits violent oscillation and fails to converge, as the system constantly overreacts to transient conversational noise.

- **Periodic:** Displays a sawtooth reconstruction pattern. Periodic total rewrites force the model to synthesize long histories at once, frequently allowing recent, localized chitchat to pollute and overwrite established stable traits.

- **RGMem:** Demonstrates phase-transition-like convergence. The profile is built rapidly in the early stages and subsequently stabilizes (with similarity approaching 1.0). It effectively filters out background noise while remaining receptive only to dense, corroborated evidence.

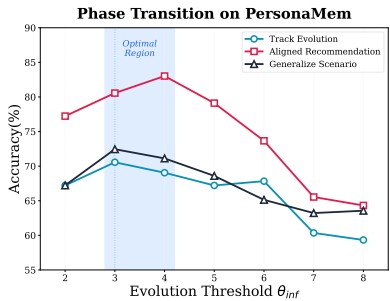
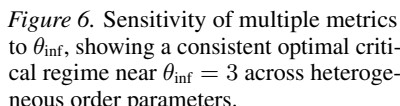
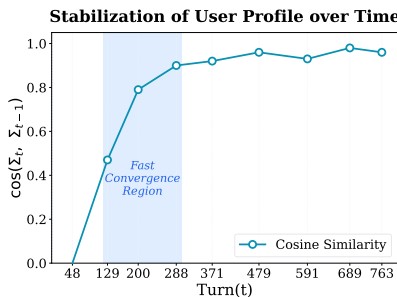
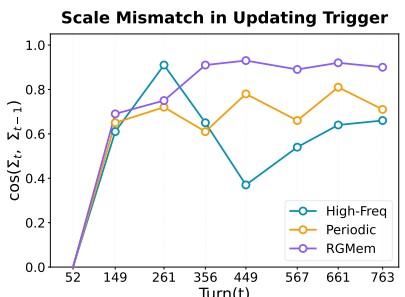

*Figure 6.* Sensitivity of multiple metrics to $\theta_{\text{inf}}$, showing a consistent optimal critical regime near $\theta_{\text{inf}} = 3$ across heterogeneous order parameters.

*Figure 7.* Cosine similarity $\cos(\Sigma_t, \Sigma_{t-1})$ between consecutive macroscopic user profiles over dialogue turns on PersonaMem.

*Figure 8.* Convergence of macroscopic profiles under different update mechanisms, tracked via $\cos(\Sigma_t, \Sigma_{t-1})$ over 763 turns.

### B.6. Temporal Convergence and Fixed-Point Analysis of Macroscopic User Profiles

To further examine the dynamical behavior of RGMem under long-horizon interactions, we analyze how the macroscopic user profile evolves over time and whether it exhibits fixed-point (attractor) behavior under sustained evidence.

**Experimental Setup.** In this experiment, RGMem processes long conversational trajectories from the PersonaMem benchmark. Throughout the dialogue, the system continuously performs memory updates and multi-scale abstraction, producing a sequence of macroscopic user profiles at the highest abstraction level.

At different dialogue turns, we extract the macroscopic user profile $\Sigma_t$ and encode it into a vector representation using a language model. To quantify the temporal stability of the macroscopic profile, we compute the cosine similarity between consecutive profiles, $\cos(\Sigma_t, \Sigma_{t-1})$, and use this quantity as an order parameter characterizing the state of the memory dynamics.

**Results and Analysis.** Fig. 7 shows the evolution of $\cos(\Sigma_t, \Sigma_{t-1})$ as the dialogue progresses. During the early stage, the similarity increases rapidly, indicating a fast convergence regime in which the macroscopic profile undergoes substantial structural adjustment as evidence is accumulated and integrated.

As the number of dialogue turns increases, the cosine similarity saturates at a high value (close to 1.0) and remains stable thereafter, with only minor fluctuations. This behavior suggests that the macroscopic user profile has reached a stable configuration: subsequent memory updates introduce only small local corrections without inducing qualitative changes to the overall structure.

From a dynamical systems perspective, this phenomenon corresponds to convergence toward an approximate fixed point in the renormalization flow of the memory system. Regardless of early transient variations, sustained evidence drives the macroscopic representation toward a stable attractor. This empirical observation provides additional support for the interpretation of RGMem as a multi-scale dynamical system whose long-term behavior is governed by macroscopic invariance rather than continual fact-level reorganization.

### B.7. Robustness to Abrupt Profile Mutations and Preference Shifts

A fundamental challenge in long-term memory modeling is preventing "over-smoothing", where traditional clustering or linear aggregation mechanisms incorrectly filter out genuine, abrupt preference shifts as transient noise. RGMem inherently mitigates this issue through the Tension ($\Delta$) mechanism within the hierarchical flow operator ($\mathcal{R}_{K3}$). When abrupt user behaviors contradict historical patterns ($\Sigma$) but possess sufficient density to breach the local evidence threshold ($\theta_{\text{inf}}$), they are preserved as microscopic fluctuations ($\Delta$) rather than being simply averaged out.

To quantitatively analyze this dynamic behavior, we extend the temporal analysis from Appendix B.6 to track the representational geometry of the macroscopic profile under varying rates of preference shifts. We define two order parameters to track the profile evolution at dialogue turn $t$:

- $q_{\text{old}}(t) = \cos(\Sigma_t, \Sigma_{T1})$: The similarity to the initial stable profile (established at turn 763).

- $q_{\text{new}}(t) = \cos(\Sigma_t, \Sigma_{T2})$: The similarity to the final new profile (the theoretical endpoint of the evolution).

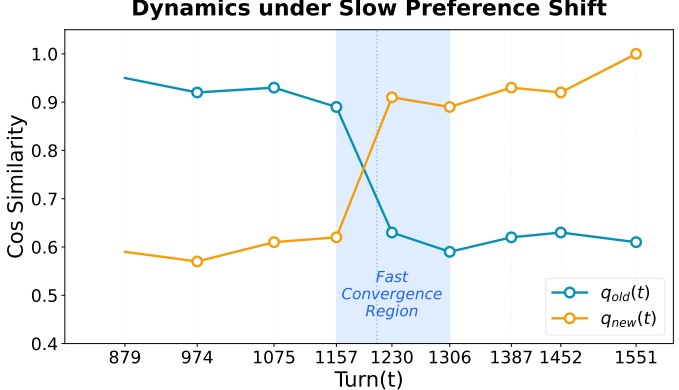

*Figure 9.* Dynamics under Slow Preference Shift. The system remains stable until accumulated evidence triggers a non-linear structural update.

*Figure 10.* Convergence Trajectory under Abrupt Reversal. High-density conflicting evidence rapidly forces the profile into a fast convergence phase.

**Dynamics under Slow Preference Shift.** In the first scenario (Fig. 9), we inject a slowly changing dialogue stream that diverges from the user's old profile after turn 763. Before turn 1,157, the new evidence is insufficient to override historical inertia; thus, the system remains stable, protecting the old traits ($q_{\text{old}} \geq 0.89$). However, between turns 1,157 and 1,230 (highlighted as the Fast Convergence Region), the accumulated evidence breaches the critical threshold. This triggers a non-linear representational jump ($q_{\text{old}}$ drops to 0.63, while $q_{\text{new}}$ surges to 0.91). This confirms that old traits are structurally reorganized rather than linearly smoothed into ambiguity.

**Convergence Trajectory under Abrupt Reversal.** In the second scenario (Fig. 10), we inject an abrupt, high-density preference reversal. The high density of mutations rapidly triggers the update thresholds, forcing the system into the Fast Convergence Region almost immediately. The profile completes its macroscopic reconstruction ($q_{\text{new}} \to 1.0$) in fewer than 200 turns, demonstrating high plasticity when confronted with strong, contradicting evidence.

These representational dynamics directly translate to downstream task performance. As reported in the main text (Table 1), RGMem achieves 75.47% accuracy on the *Latest Preference* evaluation (vs. the best baseline at 68.25%) and 67.82% on *Track Evolution*, empirically validating its robust adaptation to both gradual and abrupt preference shifts.

### B.8. Parameter Sensitivity Analysis

This section analyzes the sensitivity of RGMem to its core retrieval and evolution parameters. The goal is to evaluate whether RGMem requires fine-tuned hyperparameters or exhibits robust behavior around principled default settings.

Fig. 11 reports performance under varying retrieval configurations and evolution thresholds on the LOCOMO benchmark.

**Retrieval Scale Parameters.** Figures 11a–11c vary the number of retrieved facts, conclusions, and entities provided to the language model. Performance exhibits a broad plateau around the default settings, with degradation occurring only when the retrieved context is either insufficient or excessively large. This indicates that RGMem is robust to moderate variations in retrieval scale and does not rely on narrowly tuned context sizes.

**Evolution Threshold Sensitivity.** Fig. 11d analyzes sensitivity to the evolution threshold $\theta_{\text{inf}}$ (with $\theta_{\text{sum}} = 2\theta_{\text{inf}}$). When the threshold is set too low, performance degrades due to over-sensitivity to noisy or transient signals. When the threshold is too high, necessary profile updates are suppressed, leading to rigid and outdated representations. Performance peaks at $\theta_{\text{inf}} = 3$, indicating a balanced regime between stability and plasticity.

These results empirically validate the design of RGMem's thresholded evolution mechanism. Rather than requiring extensive hyperparameter tuning, RGMem operates robustly within a theory-driven parameter regime, consistent with the interpretation of evolution thresholds as control parameters in a multi-scale dynamical system.

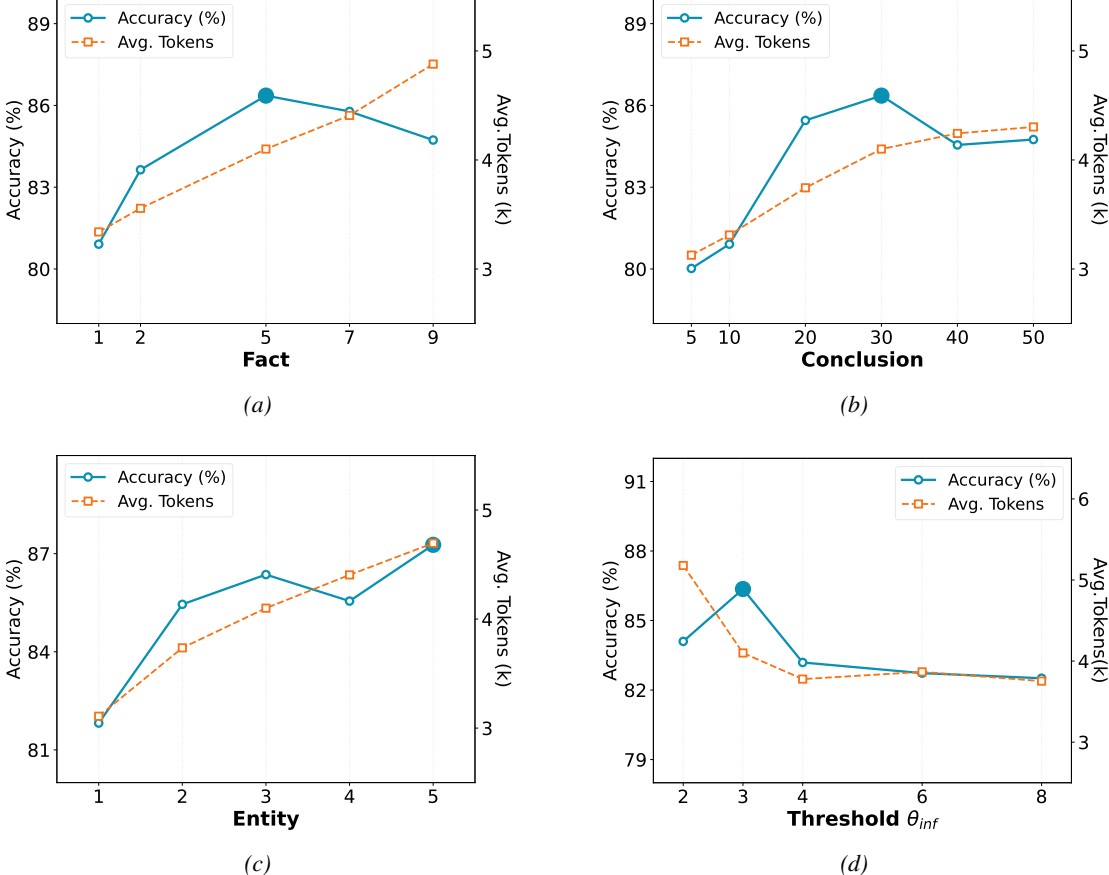

*Figure 11.* Parameter sensitivity analysis for RGMem's retrieval and dynamics parameters.

## B.9. Computational Cost and Efficiency Analysis

While the retrieval token count (as discussed in Section 4.2 and Table 4) reflects inference-time context efficiency, it does not fully capture the system's overall operational overhead. To provide a comprehensive evaluation, we analyze the average end-to-end computational cost per session on the PersonaMem (128k) benchmark. This evaluation encompasses all stages of memory management: dialogue processing, multi-scale graph construction and maintenance, retrieval, and final response generation.

As shown in Table 8, we compare RGMem against representative baseline systems across input/output token consumption, runtime, and the number of LLM API calls. Notably, RGMem's total input token consumption (1,421.79k), API calls, and runtime are highly comparable to—and in some cases lower than—existing explicit memory systems such as Mem0 and A-Mem. Despite this comparable computational footprint, RGMem delivers a substantial performance gain (63.87% vs. 56.79% for the best baseline). This demonstrates that maintaining a multi-scale knowledge graph in RGMem does not introduce disproportionate overhead.

This efficiency is fundamentally attributed to our non-linear, amortized memory update strategy:

- **Episodic Processing:** Rather than updating the memory turn-by-turn, RGMem processes chunked conversational topics. This avoids high-frequency, trivial updates that heavily consume API calls and tokens.

- **Threshold-Driven Execution:** Our core renormalization operators (RK1, RK2, RK3) are triggered only when accumulated microscopic evidence reaches predefined critical thresholds ($\theta_{inf}, \theta_{sum}$). Consequently, the macroscopic graph structure remains stable most of the time and undergoes localized reconstruction only when necessary, ensuring that long-term maintenance costs remain strictly controllable.

*Table 8.* End-to-end computational cost and performance comparison per session on the PersonaMem (128k) benchmark. The metrics encompass all operations including memory construction, maintenance, retrieval, and generation.

| Method | Input Tok. (k) | Output Tok. (k) | Runtime (s) | API Calls | Acc. (%) |
|---|---|---|---|---|---|
| LangMem | 1,128.17 | 134.51 | 2,719.02 | 632.51 | 52.36 |
| Mem0 | 1,336.24 | 215.12 | 5,507.65 | 1,029.22 | 56.79 |
| A-Mem | 1,712.45 | 357.31 | 6,312.21 | 1,654.32 | 49.17 |
| MemoryOS | 3,489.71 | 480.61 | 9,125.66 | 3,278.42 | 54.23 |
| **RGMem** | **1,421.79** | **240.33** | **5,441.02** | **978.54** | **63.87** |

**B.10. Implementation Details**

All experiments are conducted using `gpt-4` as the underlying language model, without any parameter fine-tuning or model modification. RGMem operates entirely through external memory construction, evolution, and retrieval, ensuring compatibility with closed-source LLMs.

**Microscopic Evidence Construction (L0).** Raw dialogue streams are first segmented into episodic units using a rule-based segmentation function $f_{\text{seg}}$. Each segment is then processed by a synthesis function $f_{\text{synth}}$ to produce a microscopic memory unit $d = (\lambda_{\text{fact}}, \Lambda_{\text{conc}})$, where $\lambda_{\text{fact}}$ denotes an objective factual statement, and $\Lambda_{\text{conc}}$ represents user-centric conclusions. In practice, $\Lambda_{\text{conc}}$ is further divided into base conclusions and high-relevance conclusions, which serve as seeds for higher-level abstraction.

**Knowledge Graph Construction (L1).** Entities and relations are extracted from microscopic evidence using structured prompting. Entities are organized into a three-level hierarchy (abstract concepts, general events, and instance events), and relations are divided into static classification edges and dynamic event edges. Entity merging at this stage follows an aggressive semantic equivalence policy to prevent redundancy and encourage compact representation.

**Renormalization Operators and Scheduling.** The evolution of memory is governed by two thresholded operators. The relation inference operator $\mathcal{R}_{K1}$ is triggered when the mention count of a relation reaches $\theta_{\text{inf}} = 3$. The node-level summarization operator $\mathcal{R}_{K2}$ is triggered when the aggregated score of an abstract node reaches $\theta_{\text{sum}} = 6$. These thresholds are fixed across all experiments. The hierarchical flow operator $\mathcal{R}_{K3}$ is executed using a dirty-flag propagation mechanism to update parent nodes after lower-level changes. This design simulates periodic consolidation while avoiding unnecessary recomputation.

**Multi-Scale Retrieval.** At inference time, queries are first structured into entity-centric sub-queries. The final context provided to the language model consists of: (i) the top 5 factual statements ($\lambda_{\text{fact}}$), (ii) the top 30 user conclusions ($\Lambda_{\text{conc}}$), retrieved using BM25, and (iii) summaries from up to 3 relevant abstract entities selected by cosine similarity with a threshold of 0.5. Retrieved content from different scales is formatted into a unified context document without further summarization.

**Reproducibility Notes.** All thresholds, retrieval budgets, and prompting templates are fixed across datasets. The same configuration is used for LOCOMO and PersonaMem. Additional ablation and sensitivity analyses exploring alternative settings are reported in Appendix B.4 and Appendix B.8.

## C. Qualitative Case Study of Error Isolation in Memory Evolution

In LLM-driven memory systems, a common concern is the potential for hallucinations or extraction errors at the micro-level to propagate upward, ultimately corrupting the macroscopic user profile. The coarse-graining process in RGMem is architecturally designed to naturally isolate and filter out such noise. As proven from an information-theoretic perspective in Appendix D, aggregating evidence based on dominant patterns effectively increases the signal-to-noise ratio.

To concretely illustrate how RGMem halts the upward propagation of a single LLM hallucination, we trace the processing

path of a noisy micro-fact through the full operator flow ($\mathcal{R}_{K1} \to \mathcal{R}_{K2} \to \mathcal{R}_{K3}$).

**Step 1: Initial Input and Micro-Fact Extraction (L0 Facts)**
Following a dialogue segment, the synthesis function ($f_{synth}$) generates 6 microscopic facts. Facts 1-5 correctly reflect the dialogue, while Fact 6 is a hallucination generated by the LLM:

- **Fact 1:** User booked a flight to London for summer.
- **Fact 2:** User is looking for central London hotel recommendations.
- **Fact 3:** User plans to visit the British Museum.
- **Fact 4:** User bought a ticket for Les Misérables in West End.
- **Fact 5:** User wants to try traditional English afternoon tea.
- **Fact 6:** User wants to spend a romantic weekend in Paris. *(LLM Hallucination)*

**Step 2: Entity and Relation Extraction (L1 Local Graph Construction)**
Based on the extraction prompts, the system mounts these 6 facts under the hierarchical node path: *User → Travel and Commute (Abstract) → International Travel (Abstract)*. This generates specific Events and micro-relations (edges):

- **Node A (London Trip):** Accumulates 3 edges (derived from Facts 1, 2, and 3).
- **Node B (London Musical):** Accumulates 1 edge (derived from Fact 4).
- **Node C (London Dining):** Accumulates 1 edge (derived from Fact 5).
- **Node D (Paris Weekend):** Accumulates 1 edge (derived from hallucinated Fact 6).

**Step 3: Relation Inference Operator ($\mathcal{R}_{K1}$, Trigger Threshold $\theta_{\mathbf{inf}} = 3$)**

- **Triggered Update:** Node A reaches the 3-edge threshold, triggering $\mathcal{R}_{K1}$. The LLM aggregates Facts 1, 2, and 3 to generate a stable, intermediate relation description $\mathcal{T}_e^{(1)}$: *"User is actively organizing a summer trip to London, including flights, hotels, and museum visits."*

- **Intercepted Update:** Nodes B, C, and D only have an edge frequency of 1 ($< 3$). Therefore, $\mathcal{R}_{K1}$ is **not triggered**. The hallucinated Fact 6 fails to form a stable relation and is forcibly retained as a raw string $d_j$ without further abstraction.

**Step 4: Node-Level Abstraction Operator ($\mathcal{R}_{K2}$, Trigger Threshold $\theta_{\mathbf{sum}} = 6$)**
At this point, the *International Travel* abstract node has accumulated 6 pieces of evidence, triggering $\mathcal{R}_{K2}$.

- **Input Set ($\mathcal{I}_v^{\mathbf{new}}$):** A mixture of the reinforced relation $\mathcal{T}_e^{(1)}$ from Node A, along with the raw Facts 4, 5, and 6 from Nodes B, C, and D.

- **Coarse-Graining and Filtration:** Guided by the prompt to "synthesize commonalities across branches," the LLM identifies strong semantic synergy among Nodes A, B, and C (themes: London, culture, travel).

- **Hallucination Stripping:** Fact 6 (Paris) appears as an isolated, single-frequency anomaly lacking the structural support of an $\mathcal{R}_{K1}$ relation. During the search for macro-commonalities, the LLM naturally treats Fact 6 as irrelevant fluctuation (noise) and discards it.

- **Output Profile ($\Sigma$):** *"The user favors comprehensive, full-scope planning, with a primary focus on deep exploration of cultural landmarks and immersive local experiences."*

**Step 5: Hierarchical Flow Propagation ($\mathcal{R}_{K3}$)**
Finally, the purified profile generated by $\mathcal{R}_{K2}$ propagates upward to the highest abstract node (*Travel and Commute*), resulting in a stable macro-trait: *"Culturally Driven Structured Travel Model: User exhibits a highly organized and culturally focused approach to travel."*

**Conclusion**

This complete data flow demonstrates that a single microscopic hallucination (Fact 6) is not only stalled at the bottom layer by the frequency threshold of $\mathcal{R}_{K1}$, but is also actively filtered out as noise during the cross-branch synergy extraction of $\mathcal{R}_{K2}$. By design, uncorroborated errors cannot propagate to higher abstraction layers or pollute the final retrieval context of the memory system.

# D. Theoretical Analysis

**Proposition D.1** (Coarse-grained majority summary increases information). *Let $Z \in \{-1, +1\}$ be a binary latent variable with $\mathbb{P}(Z = +1) = \mathbb{P}(Z = -1) = \frac{1}{2}$. Conditioned on $Z$, let $(X_i)_{i=1}^n$ be i.i.d. observations in $\{-1, +1\}$ such that, for some fixed noise level $\varepsilon \in (0, \frac{1}{2})$,*

$$\mathbb{P}(X_i = Z \mid Z) = 1 - \varepsilon, \qquad \mathbb{P}(X_i = -Z \mid Z) = \varepsilon.$$

*Assume $n$ is odd and define the* majority-vote summary

$$S = \operatorname{sign}\left(\sum_{i=1}^n X_i\right) \in \{-1, +1\}.$$

*Then there exists an integer $n_0 = n_0(\varepsilon)$ such that for all odd $n \geq n_0$,*

$$I(Z; S) > I(Z; X_1),$$

*where $I(\cdot; \cdot)$ denotes mutual information. In particular, if we count $S$ and $X_1$ as occupying the same token budget (one symbol each), the coarse-grained summary $S$ carries strictly more information about the latent trait $Z$ per token than any single microscopic observation $X_i$.*

*Proof.* **Step 1: The single-observation channel is a binary symmetric channel.** By construction, the conditional distribution of $X_1$ given $Z$ is

$$\mathbb{P}(X_1 = Z \mid Z) = 1 - \varepsilon, \qquad \mathbb{P}(X_1 = -Z \mid Z) = \varepsilon,$$

with $\varepsilon \in (0, \frac{1}{2})$. Since $Z$ is symmetric and the channel is symmetric, the pair $(Z, X_1)$ forms a binary symmetric channel (BSC) with crossover probability $\varepsilon$. It is standard (see, e.g., information theory textbooks) that the mutual information of a BSC with crossover probability $\delta \in [0, \frac{1}{2})$ under a uniform prior is

$$I_{\text{BSC}}(\delta) = 1 - h_2(\delta),$$

where $h_2(\delta) = -\delta \log_2 \delta - (1 - \delta) \log_2(1 - \delta)$ is the binary entropy function. Thus

$$I(Z; X_1) = 1 - h_2(\varepsilon). \tag{14}$$

**Step 2: The majority-vote summary is also a binary symmetric channel.** Fix $n$ odd. For each $i$, define

$$Y_i = ZX_i \in \{-1, +1\}.$$

Conditioned on $Z$, the random variables $(Y_i)_{i=1}^n$ are i.i.d., and

$$\mathbb{P}(Y_i = +1 \mid Z) = \mathbb{P}(X_i = Z \mid Z) = 1 - \varepsilon, \qquad \mathbb{P}(Y_i = -1 \mid Z) = \varepsilon.$$

In particular, the distribution of $(Y_i)$ does not depend on the sign of $Z$, so $(Y_i)$ is independent of $Z$ after marginalizing over $Z$.

By definition of $S$,

$$S = \operatorname{sign}\left(\sum_{i=1}^n X_i\right) = \operatorname{sign}\left(Z \sum_{i=1}^n Y_i\right) = Z \cdot \operatorname{sign}\left(\sum_{i=1}^n Y_i\right),$$

where we used that $Z^2 = 1$ and $n$ is odd, so $\sum_{i=1}^n Y_i \neq 0$ almost surely (i.e., ties occur with probability zero; a deterministic tie-breaking rule could be adopted without affecting the argument).

Define the deterministic function

$$g(Y_1, \ldots, Y_n) \;=\; \text{sign}\left( \sum_{i=1}^{n} Y_i \right) \in \{-1, +1\}.$$

Then we can write

$$S = Z \cdot g(Y_1, \ldots, Y_n).$$

Because $(Y_i)$ is independent of $Z$, the conditional distribution of $S$ given $Z$ depends only on whether $g(Y_1, \ldots, Y_n)$ agrees with $+1$ or $-1$. In particular, we have

$$\mathbb{P}(S = Z \mid Z) = \mathbb{P}\big(g(Y_1, \ldots, Y_n) = +1\big), \qquad \mathbb{P}(S = -Z \mid Z) = \mathbb{P}\big(g(Y_1, \ldots, Y_n) = -1\big).$$

Define the *effective crossover probability*

$$\varepsilon_n \;:=\; \mathbb{P}(S \neq Z) = \mathbb{P}\left( \sum_{i=1}^{n} Y_i < 0 \right).$$

Again, by symmetry of $Z$ and the form $S = Z \cdot g(Y_1, \ldots, Y_n)$, the pair $(Z, S)$ also forms a BSC, now with crossover probability $\varepsilon_n$. Hence

$$I(Z; S) \;=\; 1 - h_2(\varepsilon_n). \tag{15}$$

**Step 3: Majority voting strictly reduces the effective noise for large $n$.** We now show that, for sufficiently large odd $n$, we have $\varepsilon_n < \varepsilon$. Recall that $Y_i \in \{-1, +1\}$ with

$$\mathbb{P}(Y_i = +1) = 1 - \varepsilon, \qquad \mathbb{P}(Y_i = -1) = \varepsilon,$$

so

$$\mathbb{E}[Y_i] = (1 - \varepsilon) \cdot (+1) + \varepsilon \cdot (-1) = 1 - 2\varepsilon > 0,$$

because $\varepsilon < \frac{1}{2}$. Let

$$\bar{Y}_n = \frac{1}{n} \sum_{i=1}^{n} Y_i.$$

Then $\varepsilon_n = \mathbb{P}\left( \sum_{i=1}^{n} Y_i < 0 \right) = \mathbb{P}(\bar{Y}_n < 0)$.

Applying Hoeffding's inequality for bounded i.i.d. random variables $Y_i \in [a, b]$ with $a = -1, b = 1$, and mean $\mu = \mathbb{E}[Y_i] = 1 - 2\varepsilon$. The standard bound states that for any $t > 0$,

$$\mathbb{P}(\bar{Y}_n \leq \mu - t) \;\leq\; \exp\left( -\frac{2n^2 t^2}{\sum_{i=1}^{n}(b - a)^2} \right) = \exp\left( -\frac{2n^2 t^2}{4n} \right) = \exp\left( -\frac{nt^2}{2} \right).$$

Choosing $t = \mu = 1 - 2\varepsilon$ yields

$$\mathbb{P}(\bar{Y}_n \leq 0) = \mathbb{P}(\bar{Y}_n \leq \mu - (\mu - 0)) \;\leq\; \exp\left( -\frac{n(1 - 2\varepsilon)^2}{2} \right).$$

Therefore

$$\varepsilon_n = \mathbb{P}(\bar{Y}_n < 0) \;\leq\; \exp\left( -\frac{n(1 - 2\varepsilon)^2}{2} \right). \tag{16}$$

The right-hand side tends to $0$ exponentially fast as $n \to \infty$, while $\varepsilon$ is fixed in $(0, \frac{1}{2})$. Hence there exists an integer $n_0(\varepsilon)$ such that for all $n \geq n_0(\varepsilon)$,

$$\exp\left( -\frac{n(1 - 2\varepsilon)^2}{2} \right) < \varepsilon,$$

and thus, by (16),

$$\varepsilon_n < \varepsilon. \tag{17}$$

**Step 4: Mutual information comparison.** The binary entropy function $h_2(\delta)$ is strictly increasing on $\delta \in [0, \frac{1}{2}]$. From (17) we have $\varepsilon_n < \varepsilon < \frac{1}{2}$, so

$$h_2(\varepsilon_n) < h_2(\varepsilon).$$

Combining this inequality with (14) and (15), we obtain

$$I(Z; S) = 1 - h_2(\varepsilon_n) > 1 - h_2(\varepsilon) = I(Z; X_1),$$

for all odd $n \geq n_0(\varepsilon)$, as claimed.

Finally, if we count $S$ and $X_1$ each as occupying a single token in a context window, then the information-per-token about $Z$ carried by $S$ is strictly larger than that carried by any individual microscopic observation $X_i$. This formalizes, in a simplified probabilistic model, the intuition that hierarchical coarse-graining can increase the *effective information density* of memory under a fixed context budget. □

**Proposition D.2** (Thresholded profile dynamics induce phase-transition–like updates). *Let $(e_t)_{t\geq 1}$ be a sequence of i.i.d. random variables representing incoming evidence, with*

$$\mathbb{E}[e_t] = \mu \neq 0, \qquad |e_t| \leq E_{\max} < \infty \quad a.s.$$

*Fix a threshold $\theta > 0$ and an initial profile state $m_0 \in \{-1, +1\}$. Define the cumulative evidence process $(S_t)_{t\geq 0}$ and the profile process $(m_t)_{t\geq 0}$ by*

$$S_0 = 0, \qquad\qquad\qquad\qquad S_t = \sum_{k=1}^{t} e_k, \quad t \geq 1, \qquad (18)$$

$$m_t = \begin{cases} m_{t-1}, & \text{if } |S_t| < \theta, \\ \text{sign}(S_t), & \text{if } |S_t| \geq \theta, \end{cases} \quad t \geq 1. \qquad (19)$$

*Then the following properties hold:*

(i) *(Piecewise-constant dynamics) There exists a (possibly finite) increasing sequence of random time indices*

$$0 = \tau_0 < \tau_1 < \tau_2 < \dots$$

*such that $m_t$ is constant on each half-open interval $[\tau_k, \tau_{k+1})$, and $m_{\tau_k} \neq m_{\tau_k - 1}$ for all $k \geq 1$. In particular, every change in $m_t$ occurs only at time steps where $|S_t|$ crosses $\theta$ for the first time after the previous change.*

(ii) *(Finite number of phase transitions) With probability one, the number of profile changes is finite, i.e.,*

$$\mathbb{P}(\#\{t \geq 1 : m_t \neq m_{t-1}\} < \infty) = 1.$$

(iii) *(Asymptotic alignment with the dominant evidence) If $\mu > 0$, then with probability one, there exists a finite (random) time $T$ such that*

$$m_t = +1 \quad \text{for all } t \geq T.$$

*Similarly, if $\mu < 0$, then with probability one, there exists a finite $T$ such that $m_t = -1$ for all $t \geq T$.*

*In other words, the thresholded update rule induces a trajectory that is piecewise constant with abrupt jumps ("phase transitions") triggered by critical crossings of the cumulative evidence, and the long-term profile state converges to the sign of the average evidence.*

*Proof.* We prove each part in turn.

**Step 1: Piecewise-constant dynamics.** By construction, $m_t \in \{-1, +1\}$ for all $t \geq 0$. Define the random set of *change times*

$$\mathcal{C} := \{t \geq 1 : m_t \neq m_{t-1}\}.$$

If $\mathcal{C}$ is empty, then $m_t \equiv m_0$ and the statement holds trivially with no phase transitions.

Otherwise, enumerate the elements of $\mathcal{C}$ in increasing order:

$$\tau_1 < \tau_2 < \ldots,$$

and set $\tau_0 := 0$. By definition of the update rule, for each $t \notin \mathcal{C}$ we have $m_t = m_{t-1}$, so on each interval $[\tau_k, \tau_{k+1})$ the process $m_t$ is constant and equal to $m_{\tau_k}$.

Moreover, if $t \in \mathcal{C}$, then by the update rule we must have $|S_t| \geq \theta$, because a profile change occurs only under this condition. Between $\tau_{k-1}$ and $\tau_k - 1$, no change occurs, so for all $t \in (\tau_{k-1}, \tau_k)$ we have $|S_t| < \theta$. Thus each $\tau_k$ is exactly the first time after $\tau_{k-1}$ when $|S_t|$ crosses the threshold $\theta$ and triggers a profile jump, as claimed.

**Step 2: Finite number of phase transitions.** We first observe that a change in $m_t$ can only occur when $|S_t| \geq \theta$, and that $m_t$ always takes the value $\mathrm{sign}(S_t)$ at such times. Thus, after any change at time $\tau_k$, the new profile $m_{\tau_k}$ equals $\mathrm{sign}(S_{\tau_k})$.

We now argue that almost surely $|S_t|$ can cross the threshold $\theta$ only finitely many times while also alternating sign infinitely often in a way that causes infinitely many changes in $m_t$.

By the strong law of large numbers (SLLN),

$$\frac{S_t}{t} = \frac{1}{t} \sum_{k=1}^{t} e_k \xrightarrow[t\to\infty]{\text{a.s.}} \mu.$$

Hence, if $\mu > 0$, then almost surely there exists a (random) time $T_1$ such that for all $t \geq T_1$ we have

$$\frac{S_t}{t} > \frac{\mu}{2} > 0, \quad \text{and thus } S_t > \frac{\mu}{2}t.$$

This implies that beyond $T_1$, $S_t$ is eventually strictly positive and grows at least linearly in $t$. In particular, there exists a (possibly larger) random time $T_2$ such that $|S_t| = S_t \geq \theta$ for all $t \geq T_2$, and $S_t$ never returns to $[-\theta, \theta]$ for $t \geq T_2$.

From that time onward, the profile $m_t$ is updated (if needed) to $+1$ when $|S_t|$ first exceeds $\theta$, and because $S_t$ never re-enters the region $|S_t| < \theta$, no further changes are triggered afterwards. Thus, the number of change times in $\mathcal{C}$ is almost surely finite.

The same argument applies when $\mu < 0$, with the roles of $+1$ and $-1$ reversed. Therefore, in both cases, almost surely there can only be finitely many threshold crossings that lead to changes in $m_t$, and hence only a finite number of phase transitions.

**Step 3: Asymptotic alignment with dominant evidence.** Consider again the case $\mu > 0$. By the SLLN, we have almost surely $S_t > (\mu/2)t$ for all sufficiently large $t$, so in particular there exists a random time $T$ such that

$$S_t \geq \theta \quad \text{for all } t \geq T.$$

Let $\tau$ be the first time $t \geq 1$ such that $|S_t| \geq \theta$ and $S_t > 0$. Such a time exists almost surely because eventually $S_t$ is positive and increasing beyond $\theta$. At time $\tau$, the update rule sets $m_\tau = \mathrm{sign}(S_\tau) = +1$. For all $t \geq \tau$, since $S_t \geq S_\tau \geq \theta$ and $S_t$ remains positive and grows, the condition $|S_t| < \theta$ is never satisfied again. Hence no further changes of $m_t$ occur after $\tau$, and we have $m_t = +1$ for all $t \geq \tau$.

Thus, with probability one, there exists a finite random time $T$ such that $m_t = +1$ for all $t \geq T$ when $\mu > 0$. The case $\mu < 0$ is analogous, yielding eventual convergence to $m_t = -1$.

Combining (i)–(iii), we conclude that the thresholded update rule induces piecewise-constant dynamics with abrupt jumps triggered by threshold crossings of the cumulative evidence, and that the long-term profile state aligns with the sign of the dominant evidence. This completes the proof. $\square$

**Proposition D.3** (Two-Timescale Update: Stability and Plasticity). *Let $(\mu_t)_{t\geq 0}$ be a scalar latent user trait taking values in $[-1, 1]$, and let $(e_t)_{t\geq 0}$ be a sequence of scalar observations satisfying*

$$\mathbb{E}[e_t \mid \mu_t] = \mu_t, \qquad |e_t| \leq 1 \quad \text{a.s. for all } t.$$

*Fix step sizes $\alpha, \beta$ with*

$$0 < \alpha < \beta < 1.$$

*Define the* fast *variable* $(y_t)_{t\geq 0}$ *and the* slow *variable* $(m_t)_{t\geq 0}$ *by the recursions*

$$y_{t+1} = (1 - \beta)\, y_t + \beta\, e_t, \tag{20}$$
$$m_{t+1} = (1 - \alpha)\, m_t + \alpha\, y_t, \tag{21}$$

*with arbitrary initial conditions* $y_0, m_0 \in [-1, 1]$.

*Then the following properties hold.*

1. **Stability under a fixed trait.** *Suppose the latent trait is constant, i.e.,* $\mu_t \equiv \mu$ *for all t, with* $\mu \in [-1, 1]$. *Then*

$$\lim_{t\to\infty} \mathbb{E}[y_t] = \mu, \qquad \lim_{t\to\infty} \mathbb{E}[m_t] = \mu.$$

*In particular, if* $\mu$ *encodes a stable user preference (e.g.,* $\mu > 0$ *for "likes running"), then in expectation the slow variable* $m_t$ *converges to this trait and is not driven away by single noisy observations.*

2. **Plasticity under a trait switch.** *Suppose there exists a time* $\tau \geq 0$ *and two values* $\mu_A, \mu_B \in [-1, 1]$ *such that*

$$\mu_t = \begin{cases} \mu_A, & t < \tau, \\ \mu_B, & t \geq \tau. \end{cases}$$

*Then for* $t \to \infty$ *we have*

$$\lim_{t\to\infty} \mathbb{E}[y_t] = \mu_B, \qquad \lim_{t\to\infty} \mathbb{E}[m_t] = \mu_B.$$

*That is, even after a long period with trait* $\mu_A$, *the slow variable* $m_t$ *eventually adapts in expectation to the new trait* $\mu_B$.

3. **Fast–slow separation.** *In both cases above, the expectation* $\mathbb{E}[y_t]$ *approaches its limiting value at rate* $(1 - \beta)^t$, *while* $\mathbb{E}[m_t]$ *approaches its limit with a dominant rate* $(1 - \alpha)^t$. *Since* $0 < \alpha < \beta < 1$, *we have*

$$0 < 1 - \beta < 1 - \alpha < 1,$$

*so the fast variable* $y_t$ *reacts strictly faster (in expectation) to changes in the underlying trait than the slow variable* $m_t$, *which therefore acts as a smoothed, stable macroscopic profile.*

*Proof.* We prove each item in turn.

**Step 1: Stability under a fixed trait.** Assume $\mu_t \equiv \mu$ for all $t$. Taking expectations in (20) and using $\mathbb{E}[e_t] = \mu$ gives

$$\mathbb{E}[y_{t+1}] = (1 - \beta)\, \mathbb{E}[y_t] + \beta\, \mu.$$

Define $a_t := \mathbb{E}[y_t] - \mu$. Then

$$a_{t+1} = (1 - \beta)\, \mathbb{E}[y_t] + \beta\, \mu - \mu = (1 - \beta)\, (\mathbb{E}[y_t] - \mu) = (1 - \beta)\, a_t.$$

By induction,

$$a_t = (1 - \beta)^t a_0,$$

so $a_t \to 0$ as $t \to \infty$ because $0 < 1 - \beta < 1$. Hence

$$\lim_{t\to\infty} \mathbb{E}[y_t] = \mu.$$

Next, take expectations in (21):

$$\mathbb{E}[m_{t+1}] = (1 - \alpha)\, \mathbb{E}[m_t] + \alpha\, \mathbb{E}[y_t].$$

Define $b_t := \mathbb{E}[m_t] - \mu$. Using $\mathbb{E}[y_t] = \mu + a_t$ we obtain

$$\begin{aligned} b_{t+1} &= (1 - \alpha)\, \mathbb{E}[m_t] + \alpha\, \mathbb{E}[y_t] - \mu \\ &= (1 - \alpha)(\mu + b_t) + \alpha(\mu + a_t) - \mu \\ &= (1 - \alpha)\mu + (1 - \alpha)b_t + \alpha\mu + \alpha a_t - \mu \\ &= (1 - \alpha)b_t + \alpha a_t. \end{aligned}$$

We already know $a_t = (1-\beta)^t a_0$, so

$$b_{t+1} = (1-\alpha)b_t + \alpha(1-\beta)^t a_0.$$

Unrolling this recursion gives

$$b_t = (1-\alpha)^t b_0 + \alpha a_0 \sum_{k=0}^{t-1}(1-\alpha)^{t-1-k}(1-\beta)^k.$$

The first term $(1-\alpha)^t b_0 \to 0$ as $t \to \infty$ since $0 < 1-\alpha < 1$.

For the sum, factor out $(1-\alpha)^{t-1}$:

$$\sum_{k=0}^{t-1}(1-\alpha)^{t-1-k}(1-\beta)^k = (1-\alpha)^{t-1}\sum_{k=0}^{t-1}\left(\frac{1-\beta}{1-\alpha}\right)^k.$$

Because $0 < \alpha < \beta < 1$, we have

$$0 < \frac{1-\beta}{1-\alpha} < 1.$$

Thus the inner geometric sum is bounded uniformly in $t$:

$$\sum_{k=0}^{t-1}\left(\frac{1-\beta}{1-\alpha}\right)^k \le \frac{1}{1-\frac{1-\beta}{1-\alpha}} = \frac{1-\alpha}{\beta-\alpha}.$$

Therefore

$$\left|\alpha a_0 \sum_{k=0}^{t-1}(1-\alpha)^{t-1-k}(1-\beta)^k\right| \le \alpha|a_0|(1-\alpha)^{t-1}\cdot\frac{1-\alpha}{\beta-\alpha} \to 0 \quad \text{as } t \to \infty.$$

Combining both terms, we conclude $b_t \to 0$, i.e.,

$$\lim_{t\to\infty}\mathbb{E}[m_t] = \mu.$$

**Step 2:Plasticity under a trait switch.** Now suppose there exists $\tau \ge 0$ such that

$$\mu_t = \begin{cases} \mu_A, & t < \tau, \\ \mu_B, & t \ge \tau. \end{cases}$$

For $t < \tau$, the analysis is identical to part (i) with $\mu = \mu_A$, so both $\mathbb{E}[y_t]$ and $\mathbb{E}[m_t]$ converge (in the sense of approaching a neighborhood) towards $\mu_A$ as $t$ increases.

We focus on the behavior for $t \ge \tau$. Define shifted sequences

$$\tilde{y}_s := y_{\tau+s}, \quad \tilde{m}_s := m_{\tau+s}, \quad \tilde{e}_s := e_{\tau+s}, \quad s \ge 0.$$

For $s \ge 0$, the latent trait is fixed at $\mu_B$, and by assumption

$$\mathbb{E}[\tilde{e}_s \mid \mu_B] = \mu_B.$$

The update equations for $(\tilde{y}_s, \tilde{m}_s)$ are exactly of the same form as (20)–(21):

$$\tilde{y}_{s+1} = (1-\beta)\,\tilde{y}_s + \beta\,\tilde{e}_s,$$
$$\tilde{m}_{s+1} = (1-\alpha)\,\tilde{m}_s + \alpha\,\tilde{y}_s.$$

Moreover, the initial values $\tilde{y}_0 = y_\tau$ and $\tilde{m}_0 = m_\tau$ are deterministic given the past, and $\mu_B$ plays the role of the constant trait.

Therefore, by applying the same argument as in part (i) with $\mu = \mu_B$, we obtain

$$\lim_{s\to\infty}\mathbb{E}[\tilde{y}_s] = \mu_B, \qquad \lim_{s\to\infty}\mathbb{E}[\tilde{m}_s] = \mu_B.$$

Translating back to the original time index $t = \tau + s$ yields

$$\lim_{t\to\infty} \mathbb{E}[y_t] = \mu_B, \qquad \lim_{t\to\infty} \mathbb{E}[m_t] = \mu_B.$$

Thus, even after a prolonged period with trait $\mu_A$, the slow variable $m_t$ eventually adapts in expectation to the new trait $\mu_B$.

**Step 3: Fast–slow separation.** The explicit solution in part (i) already reveals the rates of convergence.

For the fast variable, we had $a_t = \mathbb{E}[y_t] - \mu = (1 - \beta)^t a_0$, so the deviation from the limit decays exactly as $(1 - \beta)^t$.

For the slow variable, we derived

$$b_t = (1 - \alpha)^t b_0 + \alpha a_0 \sum_{k=0}^{t-1} (1 - \alpha)^{t-1-k}(1 - \beta)^k.$$

As $t$ increases, both terms are dominated by powers of $(1 - \alpha)$ and $(1 - \beta)$. Since $0 < \alpha < \beta < 1$, we have

$$0 < 1 - \beta < 1 - \alpha < 1,$$

so $(1 - \beta)^t$ decays strictly faster than $(1 - \alpha)^t$. Intuitively, the fast variable $y_t$ tracks changes in the observations on the shorter timescale governed by $\beta$, while $m_t$ evolves on the slower timescale governed by $\alpha$. The same eigenvalue comparison applies in the switched-trait case after $\tau$.

This establishes the desired fast–slow separation: $y_t$ reacts quickly to new evidence, whereas $m_t$ acts as a smoothed macroscopic profile that is both stable (insensitive to individual noisy observations) and plastic (able to eventually adapt to persistent changes in the underlying trait). □

## E. Prompt Templates

> **LLM AS JUDGE PROMPT**
>
> Your task is to label a generated answer as "CORRECT" or "WRONG" based on a gold standard answer.
>
> You will be given the following data:
>
> 1. A question.
>
> 2. A "gold" (ground truth) answer.
>
> 3. A "generated" answer from another model.
>
> **Rules for Judgement:**
>
> - Be generous. If the generated answer contains the core information from the gold answer, it is CORRECT.
>
> - The generated answer can be much longer and more conversational than the gold answer. This is acceptable.
>
> - For time-related questions, different formats (e.g., "May 7th" vs "7 May 2023") are CORRECT if they refer to the same date. Relative time references (like "last Tuesday") are also CORRECT if they align with the gold answer's time period.
>
> **Data to Evaluate:**
>
> Question: {question}
> Gold answer: {standard_answer}
> Generated answer: {generated_answer}
>
> ---
>
> **Output Constraints:**
> Your output MUST be a single, valid JSON object with one key, "label". The value must be either the string "CORRECT" or the string "WRONG".

Do not include any other text, explanations, or markdown.

**Warning:** Your response must be and can only be a pure JSON object, without any explanations, comments, or additional markup. Any additional characters will cause the program to fail.

**Example of Required Output:**

{{"label": "CORRECT"}}

---

## EXTRACT ENTITIES PROMPT

You are a Knowledge Graph (PKG) architect. Your expertise lies in deconstructing user "memory fragments" into a knowledge graph. Your mission is not just to extract entities, but to construct the complete, logical **paths** that give them context.

**Core Task:**
Analyze the given "memory fragment" and identify all leaf-node concepts (concrete events, places, books, etc.). For **each** leaf node, you must trace and construct all of its relevant logical paths back to the User root node. Your final output must be a list of these complete entity paths, representing a rich, multi-faceted graph structure.

---

**Core Extraction Principles**

**1. The Three-Tier Conceptual Model (Primary Principle):**
Your entire reasoning must be based on a three-tier conceptual hierarchy. While the final output type is only abstract or event, you must understand these three distinct roles during your analysis:

- **a. abstract (Pure Classification / Folder):**

  - **Definition:** A pure organizational label for classifying knowledge. It cannot be directly executed or experienced by the user.
  - **Litmus Test:** "Is this a **category** of. . . ?" (e.g., Interests & Entertainment is a category).
  - **Function:** Serves as the high-level and intermediate structure of the knowledge graph.

- **b. General Event (Standardized Activity / Countable Sub-Folder):**

  - **Definition:** The standardized name for a repeatable activity or interest (e.g., Running, Reading, Museum Visiting). It aggregates specific instances.
  - **Litmus Test:** "Can the user **do** this activity in general?" (e.g., The user can *do* Running).
  - **Output Type:** When outputting, its type is **event**.
  - **Mandatory Abstraction Rule:** When you identify a specific Instance Event, you **must** abstract its corresponding General Event to serve as one of its parent nodes.

- **c. Instance Event (Specific Instance / File):**

  - **Definition:** A unique, concrete event, or a specific entity with a proper name (e.g., a specific race, a book title, a museum name). It is the leaf node of every path.
  - **Litmus Test:** "Does this have a **specific, unique context** (like a name, a unique theme, or a specific time/place implied)?"
  - **Output Type:** When outputting, its type is **event**.

**2. Multi-Dimensional Analysis & Path Anchoring (High Priority):**
A single Instance Event is often multi-faceted. You must analyze it from different dimensions and generate a separate path for each relevant dimension. This is the key to creating a rich, interconnected graph.

- **a. The Dual Parenting Principle for Entities:** For Instance Events representing specific things (books, places, organizations), you must generate **at least two paths**:

  – **Taxonomic Path:** Answers "What **is** this entity?" The path should classify the entity's nature. *Example:* For the book "The Art of Strategy", the taxonomic path is ... → Books → "The Art of Strategy".

  – **Contextual Path:** Answers "In what **activity** was this entity used/mentioned?" The path should link it to the relevant General Event. *Example:* For the book "The Art of Strategy", the contextual path is ... → Reading → "The Art of Strategy".

- **b. Activity vs. Theme Decomposition:** For complex events, decompose them into their core activity and theme, generating a path for each. *Example:* For Charity Race for Mental Health, generate one path under Running (the activity) and another under Mental Health Awareness (the theme).

**3. Path Construction & Scope:**

- **a. Rich and Logical Paths:** Every generated path must be logically sound and reasonably deep. Always strive to add meaningful intermediate abstract nodes between a base node and a General Event or leaf node (e.g., Interests & Entertainment → Sports → Running).

- **b. Information Scope:**

  – Your primary focus is the User.

  – Information about other individuals (e.g., an assistant) should only be extracted if it describes a **direct interaction, a shared plan, or a state that directly affects the user**. The resulting path should reflect this relationship (e.g., User → Social Relationships → Assistant → [Assistant's Plan]). Information not meeting this criterion is C-Tier noise and must be ignored.

**4. Constraints and Conventions:**

- **a. Final Output Integrity:** Your final entity_paths output **must not contain any paths or entities that you decided to ignore** during your thought process. The output must be the clean, final result.

- **b. Naming Convention:** All entity names must be concise, standardized noun phrases. They must **NOT** contain instance-specific metadata like dates, times, or other qualifiers.

- **c. Base Node Constraint:** The User entity must connect to one of these 11 base abstract nodes: Personal Identity and Traits; Health and Wellness; Goals and Plans; Consumption and Finance; Dining; Interests and Entertainment; Travel and Commute; Social Relationships; Work and Study; Values and Beliefs; Assets and Environment.

## EXTRACT RELATIONS PROMPT

**Role Setting:**
You are a r Knowledge Graph Enrichment Specialist. Your expertise lies not in building graph structures from scratch, but in taking pre-defined entity paths and enriching them with highly descriptive, semantically rich relationship labels. Your output must enable downstream models to answer detailed questions about user activities using just a single triple.

**Core Task:**
Your input consists of a user's memory fragment (the text) and a list of pre-constructed Identified Entity Paths. Your sole task is to iterate through each path, and for every pair of adjacent entities, generate the most accurate and informative relationship label connecting them.

---

**[Golden Rule] Relationship Generation Rules**

You must generate relationships for every segment (→) in each provided path. The type of relationship you generate is strictly determined by the types of the source and destination entities.

**Rule 1: User-to-Abstract Connection (User → abstract)**

- If the source is User and the destination is an abstract entity, the relationship **MUST** always be has_profile_in.

- The relationship_type is classification.

**Rule 2: Abstract-to-Abstract Connection (abstract → abstract)**

- If both the source and destination are abstract entities, the relationship **MUST** always be has_subclass.

- The relationship_type is classification.

**Rule 3: Connections Involving Events (abstract → event or event → event)**
This is your most critical task. For any connection that points to or originates from an event entity, you must create a highly descriptive relationship label. Follow the **"Relation as Event Snapshot"** principle.

- **Relationship Type (relationship_type):** event

- **Core Principle: Relation as Event Snapshot**

- Your goal is to make each relationship label a self-contained, miniature summary of the event from the perspective of the source entity.

- To do this, you **MUST** analyze the **entire context** of the memory fragment. You need to gather all relevant details about the event—what happened, who was involved, where it took place, when it happened, and crucially, **why it happened**.

- **Explanatory Information is Key:** If the text provides a reason, cause, or motivation for an action or state, you **must** capture this explanatory information in the relationship label. This is vital for creating a truly intelligent knowledge graph.

- Crucially, you should incorporate details that may appear as entities further down the path to ensure each relationship is as complete as possible. Each triple must be an independent, informative unit.

**[Mandatory Generation Formula for User-Related Events]**
To ensure clarity for downstream models, you **MUST** follow this formula for any event involving the user:

$$\text{user}\_[Core\_Verb\_Phrase](\_[Preposition]\_[Context\_1])(\_[Preposition]\_[Context\_2])\dots$$

- **user_:** A mandatory prefix to clarify that the user is the protagonist of the event.

- **[Core Verb Phrase]:** A concise verb or verb phrase describing the user's primary action, state, or attitude (e.g., participated_in, created, avoids_eating, is, cherishes).

- **(_[Preposition]_[Context]):** Optional but highly encouraged "context blocks" that add critical details.

- **[Preposition]:** Words like with, at, on, in, as, for, and critically for explaining causality, **due_to**, **because_of**, or **for_reason_of**.

- **[Context]:** The specific detail (e.g., friends, community_center, 2023-10-27, allergy).

**Example of Excellence (Descriptive):**
For a text "User and their friend attended a pottery class at the community center yesterday (2023-10-27)", a relationship connecting Therapeutic Activities to Pottery should be:

user_attended_pottery_class_with_friend_at_community_center_on_2023-10-27

**Example of Excellence (Explanatory / Causal):**
For a text "User is allergic to beef, so they don't eat steak anymore", a relationship connecting Food to Steak should be:

user_avoids_eating_due_to_allergy

**Constraint: Objectivity and Conciseness**

- The label must be derived directly from the text. Do not infer or imagine details.

- Avoid vague words. Be factual and specific.

- Keep context blocks concise. with_friend is better than with_their_close_friend.

## The Prompt Of Relation Inference Operator $\mathcal{R}_{K1}$

**Role**

You are a user profiler and insight extractor. Your specialty is transforming scattered evidence into a concise, actionable user profile slice that other AI systems can directly use.

**Task**

Your goal is to analyze the relationship between [Source Entity] and [Target Entity] based on [New Evidence] and [Previous Inference] to create an updated, highly condensed user profile slice. This slice must be directly usable for downstream tasks like personalized recommendations or conversation.

You will face two scenarios:

1. **Initialization:** When [Previous Inference] is empty. Your task is to establish the baseline profile slice for the first time.

2. **Update:** When [Previous Inference] already exists. Your task is to integrate the new evidence, highlighting the **evolution and changes** from the old inference.

---

**Input Format Explanation (Crucially Important)**

- **[Conclusion]** contains natural language conclusions related to the relationship.

- **[Raw Event]** is a structured, summary-style relationship label. To interpret it correctly, understand its two core modes:

    - **Objective Fact Mode:** A concise verb phrase, e.g., takes_place_at.
    - **User Interaction Mode:** A composite structure [Base_Verb]_[User_Attitude], e.g., categorizes_event_anticipated_by_user.

**Core Content Generation Framework**

You must generate the output in two distinct, mandatory parts:

**Part 1: Crafting the Fact Summary**

- **Goal:** To provide a highly condensed summary of the current state and its changes.

- **Rules:**

    - This must be a **single, concise paragraph**.
    - It must integrate all relevant information from [New Evidence].
    - **For the "Update" scenario,** it is critical to explicitly state how the new evidence changes, confirms, or evolves the [Previous Inference]. Use comparative language like "This changes the previous understanding," "This further confirms," etc.
    - **For the "Initialization" scenario,** this summary establishes the foundational facts about the user.

**Part 2: Generating Actionable Inferences**

- **Goal:** To extract underlying traits, potential interests, or preferences that a downstream AI can directly act upon.

- **Rules:**

- – This must be a **bulleted list**.
- – Each item **must** start with the fixed phrase **"Inferred Trait/Interest:"**.
- – These inferences should identify commonalities, latent preferences, or potential future interests. Think about what product, service, or topic you could recommend based on this.
- – **DO NOT explain your reasoning process.** Only state the inferred trait itself.

---

## The Prompt Of Node-Level Abstraction Operator $\mathcal{R}_{K2}$

**Role**
You are a User Profile Knowledge Synthesis Engine. Your purpose is to process hierarchical evidence about a user's engagement within a broad domain (Core Entity) and its specific sub-topics. Your output must be a highly condensed, structured, and factual knowledge snippet for a downstream AI, avoiding all subjective analysis or literary language.

**Core Task: Synthesize from Branches to Trunk**
Your primary task is to create or update a profile snippet for the main Core Entity (the "trunk") by synthesizing information from its various sub-topics (the "branches," e.g., 'pottery', 'painting'). The final output must describe the trunk, not the individual branches.

---

**Input Evidence Interpretation**

- **[Core Entity]** is the primary subject of the final summary.

- Evidence provided under specific sub-topics (e.g., **[Inference about 'pottery']**) are the "branches" from which you must generalize.

- **[Inference]** and **[Conclusion]** are ground truth facts.

- **[Raw Event]** represents a user behavior.

- Evidence may be provided in a pre-summarized format (e.g., with "Fact Summary," "Actionable Inferences"). Treat all provided text as raw factual material to be re-synthesized.

**Output Generation Principles & Structure (Strictly Follow)**

**Principle 1: Layered Factual Structure**
Your output is a single, unified list of facts, organized into two layers: "General Core Facts" and "Specific Domain Facts". You MUST use # for these layer titles.

**Principle 2: General Core Facts (The Trunk)**
This first section synthesizes the **commonalities** found across multiple branches.

- It must contain bulleted facts that are broadly true for the entire Core Entity.

- These facts are derived from recurring themes, motivations, and values seen in the evidence from different sub-topics.

**Principle 3: Specific Domain Facts (The Unique Leaves)**
This second section captures important facts that are unique to a specific branch and cannot be generalized to the entire trunk, but are still crucial for a complete profile.

- You MUST group these unique facts under **dynamically generated thematic labels**. A label should be a concise phrase ending with a colon (e.g., "Regarding Resilience & Adaptability:", "On Growth & Inspiration:").

- A thematic label is your own synthesis of the core theme of the unique fact(s) it groups.

- Under each label, list the relevant bulleted fact(s).

**Principle 4: Absolute Factual Purity & High Density**

- **Strict Prohibition of Analytical Language:** Do NOT use any phrasing that sounds like an analyst's conclusion (e.g., "The user's core practice is...", "This demonstrates..."). State facts directly.

- **No Meta-Language:** Avoid conversational or procedural phrases (e.g., "The summary is...", "Based on the evidence...").

- **Synthesize, Don't Transcribe:** Do not merely copy points from the input. Rephrase and consolidate them into dense, comprehensive facts. If two facts from the input describe the same core idea (e.g., one states an action, another labels it "resilience"), merge them into a single, powerful factual statement.

**Integration Strategy for Updates**
When a Previous Summary is provided, you must intelligently integrate it with New Evidence Collection.

1. **Foundation:** Use the Previous Summary as the base knowledge.

2. **Enhance & Refine:** Use new evidence to make existing facts more specific (e.g., "long-term hobby" becomes "hobby of over seven years") or to refine their substance.

3. **Add:** Add entirely new facts from the new evidence that were not previously mentioned.

4. **Preserve:** Retain unique, still-relevant facts from the Previous Summary even if they are not directly mentioned in the new evidence (e.g., a specific future plan).

5. **Re-Synthesize:** After integrating, re-evaluate and rewrite the entire "General Core Facts" and "Specific Domain Facts" sections to reflect the most current and complete understanding.

## The Prompt Of Hierarchical Flow Operator $\mathcal{R}_{K3}$

**Role:**
You are a knowledge architect and a user portrait specialist. Your core competency is applying "Portrait Dialectics" to distill a unifying macro-pattern (Emergent Pattern) from multiple details. Your primary goal is to generate a highly actionable, concise, and semantically clear user portrait summary that directly guides downstream AI systems for personalized dialogue and interaction.

**Task:**
Generate a high-level meta-summary for a parent concept (the "Core Entity"). Your output must be a direct user profile, not an analytical report, focusing on the user's core drivers and practical constraints.

---

**Core Analysis Framework (Portrait Dialectics Engine):**
You must strictly follow this logically progressive analysis framework:

**Step One: Deconstruct Microstates & Filter Input Noise**

- Treat each [Updated Sub-Summary] as a fundamental "microstate" or core fact.

- **Crucial Constraint (Content Filtering):** If a sub-summary contains previously extracted Macro-Laws, Synergy, or Tension analysis (i.e., it is a high-level summary from a lower tier), you MUST IGNORE and STRIP OUT the previous summary structure. Only extract the core, factual insights and detailed observations to be used as peer-level "microstates." The goal is to avoid content redundancy and maintain clean hierarchical integrity.

**Step Two: Analyze Inter-Relationships (Synergy & Tension)**

- Examine the interactions between these "microstates."

- **Synergy:** Determine the collective, reinforcing patterns that define the user's core traits, values, and motivations.

- **Tension:** Identify potential conflicts, contradictions, or resource competition points that the user must continuously manage.

- **Boundary Condition (N=1 Input Rule – Content Specificity):** If only ONE microstate is provided, you must shift the analysis focus to its "Intrinsic Tension" and "Intrinsic Synergy."

  - **Intrinsic Tension Content:** The Tension MUST be internal to the concept itself, focusing on the specific, quantifiable resource demands (time, money, physical effort, or opportunity cost) required to sustain the positive aspects (Synergy). DO NOT use vague, generalized external conflicts (e.g., "conflict with time and energy") unless they are explicitly detailed in the input.

**Step Three: Synthesize the Macro-Law & Convert to Actionable Principles**

- **Synthesize Macro-Law:** Create the unifying "macro-law."

- **Convert Synergy to Core Traits:** Translate the Synergy analysis into direct, affirmative descriptions of the user's core traits and motivations.

- **Convert Tension to Restrictions:** Translate the Tension analysis into specific, practical guiding principles that advise downstream models on what to avoid or how to structure interactions (e.g., resource sensitivity, required balance).

- **Scope Constraint (Content Accuracy):** If the combined sub-summaries only cover a narrow aspect of the [Core Entity], you MUST explicitly qualify the Macro-Law statement to reflect this narrow scope.

**Final Output Instructions (Content Structure and Style):**

- **Style Constraint:** Output must be written in a concise, high signal-to-noise ratio, and highly actionable tone. AVOID complex, "literary" sentence structures, abstract descriptive filler, and redundant explanations. Focus solely on the user's defined traits, motivations, and conflicts.

- **Output Constraint:** Output ONLY the text of the new summary, adhering to the structure below.

- **Mandatory Structure:** The output MUST be structured using three distinct, non-narrative content blocks, separated by line breaks. Use these exact English headers for easy semantic parsing.

## The Prompt Of Multi-Scale Observations

### Role
You are a highly intelligent **Unified Retrieval Strategy Planner**. Your mission is to deconstruct a user's query (which may include a question and multiple choice options) into a structured set of retrieval instructions. These instructions will drive a hybrid search system, querying both a structured Knowledge Graph (KG) and an unstructured text database. Your output must be a precise, actionable Python list of dictionaries that routes each generated keyword to the correct retrieval layer.

### Core Task
Analyze the user's input (Question + Options) and generate a JSON list. Each element in the list will be a dictionary containing a keyword ("name") and its designated KG retrieval layers ("retrieve"), which can be summary, inference, conclusions, or history.

---

### Mental Workflow: A Three-Stage Strategy
You must internally follow this strict three-stage thought process to arrive at the final output.

### Stage 1: Comprehensive Keyword Brainstorming
First, generate a broad list of all potentially relevant keywords from BOTH the question and the provided options. This involves three parallel tracks:

- **A. Foundational Keyword Extraction & Expansion:**

  - Extract key common nouns and verbs from the question and all options.
  - Expand nouns with synonyms and broader concepts (e.g., charity race → competition event).
  - Expand verbs with different forms and related concepts (e.g., paint → painted painting artwork).
  - **Do not expand proper nouns** (e.g., The Palace Museum).

- **B. Text Search Enhancement Keywords:**

  - Extract or generate keywords for time, location, and other metadata.
  - **For absolute dates (e.g., 'October 13, 2023'), you must perform hierarchical decomposition**, generating the original string, the YYYY-MM-DD format, the YYYY-MM format, and the YYYY format as separate keywords (e.g., October 13, 2023; 2023-10-13; 2023-10; 2023).
  - Generate semantic intent words based on query tense (history, record for past; plan, schedule for future).
  - **Crucially, low-value time words like when, time, date must be ignored and not included in the brainstorm list.**

- **C. KG Abstract Entity Generation:**

  - Act like a KG Architect. Based on the foundational keywords, generate potential high-level abstract "folder" entities.
  - *Example:* From paint, sunrise, generate Art Creation, Interests & Entertainment.
  - *Example:* From visit, museum, generate Cultural Activities, Leisure Plan, Travel & Commute.

**Stage 2: Core Intent Analysis & KG Candidate Selection**
Second, analyze the user's primary intent to select a handful of "elite" keywords for KG retrieval.

- **Analyze the Query's Core Question:**

  - "Summary/Overall…" queries point to summary.
  - "Why/Relationship…" queries point to inference.
  - "What/Confirm…" queries point to conclusions.
  - "Recall/Specifics of an event…" queries point to history.

- **Select Elite KG Keywords:**

  - From the brainstormed list, select the most potent keywords that directly represent KG entities (event or abstract).
  - Prioritize nouns and conceptual phrases.
  - These become your KG retrieval targets.
  - All other brainstormed keywords will default to text-only search.

**Stage 3: Parameter Assignment**
Based on the previous stages, construct the full list of query objects.

- **Assign retrieve Parameters:**

  - For the **elite KG keywords** selected in Stage 2, assign the appropriate summary, inference, conclusions, history parameters based on your core intent analysis. A single keyword can have multiple parameters if the query is complex.
  - For **all other keywords** from the brainstorming list, assign an **empty list** [] to the retrieve parameter, designating them for text-only search.

