# OpenReview forum: "RGMem: Renormalization Group–inspired Memory Evolution for Language Agents"
_ICML.cc/2026/Conference — ICML 2026 regular_

### Official Review · Reviewer_oYk6 · 2026-02-17

**Soundness:** 3
**Presentation:** 2
**Significance:** 3
**Originality:** 3
**Overall Recommendation:** 3
**Confidence:** 4

**Summary:**

This paper proposes RGMem, a memory evolution framework for LLM agents inspired by the Renormalization Group (RG) perspective from statistical physics. The key idea is to treat interaction history as a multi-scale process, where fine-grained dialogue traces are progressively “coarse-grained” into higher-level user traits and long-term preferences. The method maintains a graph-based memory structure and iteratively updates or merges memory nodes through an RG-inspired flow. Experiments on multi-session dialogue benchmarks report improvements over standard retrieval-based or fixed-memory baselines in tracking evolving user states and personalization consistency.

**Compliance With Llm Reviewing Policy:**

Affirmed.

**Final Justification:**

The rebuttal does not change my concerning, so I keep my rating.

**Key Questions For Authors:**

- Can the authors clarify what concrete algorithmic benefit the RG-inspired flow provides beyond standard hierarchical summarization or clustering-based memory compression?
- How does RGMem compare against a compute- and token-budget matched recursive summarization baseline?
- When user preferences shift abruptly rather than gradually evolve, how does the coarse-graining process avoid over-smoothing or persisting outdated “stable traits”?
- Many critical design choices are described only in the supplement. Which components are essential for the reported gains, and can the method be simplified for reproducibility?

**Limitations:**

No. The limitations section should more explicitly discuss the complexity and reproducibility burden of the approach, as well as potential failure modes under non-stationary user behavior and contradictory long-term traits.

**Strengths And Weaknesses:**

Strengths:
- Soundness: The overall memory evolution pipeline is technically coherent, and the proposed multi-scale organization of interaction history is a reasonable way to address long-term personalization beyond simple fact retrieval.
- Presentation: The paper is generally well structured, and the high-level motivation (moving from utterance-level memories to trait-level representations) is clearly communicated.
- Significance: Long-term memory management remains a key bottleneck for agentic systems, and improving consistency across extended interactions is an important and timely problem.
- Originality: Framing memory evolution through an RG-style coarse-graining lens is intellectually interesting and provides a distinctive conceptual perspective compared to standard summarization-based memory approaches.

Weaknesses:
- Soundness: While the appendix provides extensive theoretical discussion of the proposed coarse-graining dynamics, the analysis relies on idealized assumptions (e.g., stability properties of summarization operators and stationary user distributions). It remains unclear how strongly these guarantees translate to the practical behavior of the proposed memory update process in realistic non-stationary settings.
- Presentation: A substantial portion of the technical details and critical implementation choices are deferred to a very long supplement, which makes it harder to identify the core algorithmic contribution and assess reproducibility from the main paper alone.
- Significance: The method introduces a fairly complex graph-based memory evolution mechanism, and the paper does not provide sufficient evidence that this added complexity is necessary compared to simpler hierarchical summarization or clustering-based memory baselines under matched compute/token budgets.
- Originality: Although the RG framing is appealing, the actual algorithmic operations appear closely related to existing multi-level memory compression and recursive summarization techniques. The paper does not convincingly isolate what unique advantage the RG-inspired formulation provides beyond a conceptual framing.

---

> ### Author Rebuttal · Authors · 2026-03-30
>
> We thank the reviewer. To address questions on non-stationarity, matched-resource comparison, and system complexity, we provide the following clarifications and add new experiments.
>
> #### **1. Algorithmic Advantages of RGMem over Standard Hierarchical Summarization**
> Standard summarization suffers from *premature abstraction*and *majority-vote over-smoothing* .
>
> To tackle the predicament, our RGMem introduces three unique algorithmic mechanisms:
> 1. Non-linear Triggering via Critical Thresholds
> 2. Explicit Separation of Synergy and Tension ($\Sigma/\Delta$)
> 3. Convergence to Predefined Base Dimensions
>
> Due to space limits, we have detailed how these three mechanisms fundamentally differentiate RGMem from standard hierarchical summarization in  **Response to Reviewer brZr (Q2)**. Please kindly refer to that section for a comprehensive understanding.
>
> #### **2. Performance Comparison under Matched Token/Compute Budgets**
> Our **`w/o RG`** ablation (Appendix Table 4) represents the exact "standard single-scale recursive summarization baseline" . It removes threshold-based phase-transition updates, forcing routine tree-like aggregation at every turn.
>
> Empirical Data:
> | Method | Average Context Tokens | Overall Accuracy (%) |
> | :--- | :---: | :---: |
> | Standard Recursive Summarization (`w/o RG`) | 8,479 | 57.37 |
> | **Full RGMem** | **7,105** | **63.87** |
>
> *Conclusion:* `w/o RG` causes severe context redundancy (+19% token consumption). RGMem’s gains stem from higher effective information density via threshold control, not increased compute budgets.
>
> #### **3. Handling Abrupt Preference Shifts and Preventing "Over-smoothing"**
> Traditional clustering smooths out abrupt shifts as noise. RGMem solves this via the **Tension ($\Delta$) mechanism** in $R_{K3}$. When abrupt behaviors breach the local threshold ($\theta_{inf}$), they are extracted as microscopic fluctuations ($\Delta$) contradicting historical patterns ($\Sigma$), rejecting simple averaging.
>
> To quantitatively validate this, we tracked RGMem's **Representational Geometry Dynamics** (extending Fig 7). We defined two order parameters tracking profile evolution at turn $t$:
> * **$q_{old}(t) = \cos(\Sigma_t, \Sigma_{T1})$**: Similarity to the initial old profile (steady state at turn 763).
> * **$q_{new}(t) = \cos(\Sigma_t, \Sigma_{T2})$**: Similarity to the final new profile (evolution endpoint).
>
> **Experiment 1: Dynamics under Slow Preference Shift**
> After turn 763, we injected a slow-changing dialogue differing from the old profile.
> | Dialogue Turn ($t$) | 879 | 974 | 1075 | **1157** | **1230** | 1306 | 1387 | 1452 | 1551 |
> | :--- | :--- | :--- | :--- | :--- | :--- | :--- | :--- | :--- | :--- |
> | **$q_{old}(t)$** | 0.95 | 0.92 | 0.93 | **0.89** | **0.63** | 0.59 | 0.62 | 0.63 | 0.61 |
> | **$q_{new}(t)$** | 0.59 | 0.57 | 0.61 | **0.62** | **0.91** | 0.89 | 0.93 | 0.92 | 1.00 |
>
> *Analysis:* Before turn 1157, insufficient evidence kept the system stable ($q_{old} \ge 0.89$). Between 1157-1230, accumulated evidence breached the threshold, causing a non-linear jump ($q_{old} \to 0.63, q_{new} \to 0.91$). Old traits are structurally stripped, not linearly averaged.
>
> **Experiment 2: Convergence Trajectory under Abrupt and Dense Reversal**
> After turn 763, we injected an abrupt, high-density preference shift.
> | Dialogue Turn ($t$) | 787  | 812  | 832  | 865  | 891  | 934  |
> | :--- | :--- | :--- | :--- | :--- | :--- | :--- |
> | **$q_{old}(t)$** | 0.94 |  0.89 | 0.56 | 0.49 | 0.53 | 0.51 |
> | **$q_{new}(t)$** | 0.46 |  0.62 | 0.92 | 0.90 | 0.89 | 1.00 |
>
> *Analysis:* High-density mutations rapidly triggered thresholds, completing convergence ($q_{new} \to 1.0$) in <200 turns, dominating macroscopic reconstruction.
>
> **Benchmark Validation:** In Table 1, RGMem scores **75.47%** on `Acknowledge Latest Preference` (vs. baseline 68.25%) and **67.82%** on `Track Full Preference Evolution`, empirically proving robust adaptation to abrupt shifts.
>
> #### **4. Essential Components, System Complexity, and Reproducibility**
> **(1) Practical Simplification under Guidance of Complex Theory**
> While Appendix B rigorously proves the mathematical logic, RGMem's engineering implementation is highly accessible and requires NO LLM fine-tuning. The complex RG theory is minimally instantiated via:
> * Two global parameters ($\theta_{inf}=3, \theta_{sum}=6$).
> * Standardized Prompts (Appendix D) guiding off-the-shelf LLMs to extract $\Sigma/\Delta$.
> Our open-sourced code allows plug-and-play reproduction with minimal barriers.
>
> **(2) Essential Components (Ablations in Appendix C.4):**
> * **Factual Foundation (`w/o L0`):** LOCOMO accuracy plummets from 86.17% to 56.29%. Macroscopic "slow variables" must anchor on microscopic "fast variables" to prevent hallucinations.
> * **RG Evolution (`w/o RG`):** Replacing threshold-updates with uniform updates drops PersonaMem accuracy to 57.37% (with ~20% more tokens) due to premature abstraction.

---

> > ### Author Rebuttal · Reviewer_oYk6 · 2026-04-02
> >
> > I remain unconvinced about the novelty of the work.
> >
> > The proposed RG-inspired framework is presented as a key contribution, but it is still unclear what fundamentally new capability it enables beyond existing hierarchical memory or summarization approaches.
> >
> > The mechanisms described appear to be combinations of known design patterns, and the paper does not establish that the RG formulation plays a necessary role, rather than serving as a conceptual wrapper over existing techniques.
> >
> > As a result, the core contribution remains insufficiently differentiated from prior work at the algorithmic level. I therefore maintain my original recommendation.

---

> > > ### Author Response · Authors · 2026-04-07
> > >
> > > ## LLM Agent Memory Lifecycle
> > > According to references [1-3],the LLM memory experiences a 5-module lifecycle:
> > > [M1.Formation]→[M2.Structuring]→[M3.Updating Triggers]→[M4.Consolidation]→[M5.Retrieval].While M1 & M5 are mature, ultra-long dynamic sequences still challenge the evolution modules (M2–M4).We next show how the RG-inspired framework resolves these issues.
> > > ## M2:Ontological Confusion & Clustering Limitation
> > > The hierarchical memory systems(e.g.,CAM[1],SGMem[4])rely on dynamic graphs & geometric clustering(KNN,semantic similarity)for macro-abstractions.Clustering captures only surface similarity, missing causal and logical relations.Over long dialogues,the lack of priors leads to chaotic macro-nodes and semantic drift.
> > > To solve this,our RG-inspired design clearly separates micro-details(fast var)from macro-traits(slow var).Instead of unconstrained clustering, we predefine semantic nodes as Fixed Points. All micro-events are mapped to these dimensions to form macro-traits.Such a strict mapping enables to avoid semantic drift and keeps the profile stable and interpretable.
> > > ### Table R1:RGMem vs.GraphRAG(a "clustering+summary" baseline fed with RGMem-L0 outputs)on 20% PersonaMem:
> > > |Method|Recall Facts|Track Pref.Evol.|Latest Pref|
> > > |-|-|-|-|
> > > |L0+GraphRAG|79.86|51.49|49.72|
> > > |RGMem|81.65|68.77|74.11|
> > > ### Analysis:
> > > i)Recall:GraphRAG retains basic facts
> > > ii)Macro Evolution:GraphRAG's accuracy drops by ~17-25% on dynamic tracking tasks.Similarity-based clustering lacks stable anchors,leading to progressive semantic drift over long dialogues
> > > iii)RGMem Advantage:By mapping micro-events to the predefined dimensions of Fixed Points,RGMem establishes a stable semantic coordinate system,improving tracking of profile changes.
> > > ## M3:Scale Mismatch in Updating Trigger
> > > Updating macro profiles is hard due to noise and delay.Existing methods usually use rigid external clocks:**high-frequent incremental updates**(e.g.,[5],per turn)inject noise, causing oscillation and wasted compute; **periodic global updates**(e.g.,[6],daily)lag behind changes and let local chats pollute profiles.
> > > Our RG's Critical Point module inspires local evidence density trigger.RGMem updates are event-driven:triggered only when micro-facts hit a threshold($\theta_{inf}=3$),allocating computation to structurally changing parts.
> > > ### Table R2:Cost & Acc.20% PersonaMem against high-freq($\theta_{inf}=1$)and periodic(scheduled global rewrite)updates.
> > > |Update Mech.|In Tok(k)|Out Tok(k)|Acc(%)|
> > > |-|-|-|-|
> > > |High-Freq|384.23|63.09|53.72|
> > > |Periodic|237.76|39.87|48.95|
> > > |RGMem|295.17|52.62|65.32|
> > > As shown in Table R2,compared to high-freq and periodic updates,RG's threshold-based trigger action significantly achieves the best Acc.performance.
> > > ### Table R3:Profile Sim.Convergence.Tracked cosine similarity $cos(\Sigma_{t},\Sigma_{t-1})$ of user profiles pre-/post-updates over 763 turns(1.0 = stable).
> > > |Update Mech.|T52|T149|T261|T356|T449|T567|T661|T763|
> > > |-|-|-|-|-|-|-|-|-|
> > > |High-Freq|0.00|0.61|0.91|0.65|0.37|0.54|0.64|0.66|
> > > |Periodic|0.00|0.65|0.72|0.61|0.78|0.66|0.81|0.71|
> > > |RGMem|0.00|0.69|0.75|0.91|0.93|0.89|0.92|0.90|
> > > ### Analysis:
> > > As shown in Table R3,high-Freq oscillates violently(no convergence).Periodic shows sawtooth reconstruction(recent chitchat pollutes stable traits).Differently,RGMem shows Phase-Transition-like Convergence:fast build,then stable,filtering noise.
> > > ## M4:"Over-smoothing" in Multi-level Compression
> > > Existing memory systems rely on LLMs for **Recursive Summarization** to compress long chats(e.g.,[7]).Yet,LLMs favor "Majority Vote",causing **Info smoothing**.During the **sudden preference reversals**,recent conflicting signals are "Averaged away" by long-time conversation history,causing sluggishness or "selective amnesia".
> > > Our RG's **Coarse-Graining** module demands preserving **Critical-Point Fluctuations**.That is to say,our operator abandons single summaries,forcing the extraction of tuple$(\Sigma,\Delta)$,to isolate new conflicting shifts $(\Delta)$ from dominant historical traits$(\Sigma)$.
> > > ### [Case Study]
> > > For the sentence:“History shows multiple strength training records.Latest turn shows one rehab exercise due to back injury.”Traditional summary treats injury as noise,yielding a false "heavy fitness" profile.Differently,our RGMem extracts $\Sigma$(Synergy)as "heavy fitness habit" AND $\Delta$(Tension)as "current rehab need," by which the over-smoothing can be well fixed.
> > > ### Table R4:Keep/remove $\Delta$ extraction
> > > |Aggregation Mech.|Latest Pref(%)|
> > > |-|-|
> > > |w/o $\Delta$ extraction|52.58|
> > > |with $\Sigma$ & $\Delta$|74.11|
> > > From the comparative results with/without extraction,we observe without system fails to capture the latest states while conflict exists,however,RG's extraction can fix it by preventing info loss.
> > > ## References
> > > [1] CAM...NeurIPS '25
> > >
> > > [2] From RAG to Memory...ICML '25
> > >
> > > [3] LIGHTMEM...ICLR '26
> > >
> > > [4] SGMEM...
> > >
> > > [5] Enabling personalized long-term interactions...
> > >
> > > [6] MemoryBank...AAAI '24
> > >
> > > [7] Iterresearch...ICLR '26

---

### Official Review · Reviewer_brZr · 2026-03-08

**Soundness:** 3
**Presentation:** 2
**Significance:** 3
**Originality:** 2
**Overall Recommendation:** 4
**Confidence:** 3

**Summary:**

This paper presents RGMem, a hierarchical memory framework inspired by the Renormalization Group (RG) theory. Specifically, it models long-term conversations through multi-scale knowledge graphs and LLM aggregation. It achieves 7-9% improvements on both LOCOMO and PersonaMem. However, the overly introduced theory makes the paper hard to read and understand. And RG is merely a conceptual metaphor and lacks a formal mechanism, making the method actually similar to existing LLM memory approaches.

**Compliance With Llm Reviewing Policy:**

Affirmed.

**Final Justification:**

The author provides additional explanations for the terms, making it clearer. So I raised my scores. However, I still have concerns about the overly introduced theory. I would not mind if this paper gets rejected.

**Key Questions For Authors:**

1. Could you please provide more details about Figure 4? It is hard to understand through the simple title "Non-linear dependence on evolution threshold." without further explanation.

2. What is the key difference of RGMem between the existing hierachical memory methods like MIRIX [1] and LightMem [2]?

[1] MIRIX: Multi-Agent Memory System for LLM-Based Agents. Arxiv 2025.07.

[2] LightMem: Lightweight and Efficient Memory-Augmented Generation. ICLR 2026.

**Limitations:**

Yes

**Strengths And Weaknesses:**

**Strengths.**
- **Interesting attempt of combining RG theory with memory problems**.  The RG-inspired memory mechanism is conceptually interesting for long-term memory modeling.

- **Well-performing on multiple benchmarks**: RGMem achieves consistent improvements across LOCOMO (7.08%) and PersonaMem (8.98%), showcasing the effectiveness of the proposed method.

**Weaknesses.**
My main concern focuses on the combination of the RG theory part.
- **The overly introduced theory makes the whole paper hard to understand.** The RG theory introduced in this paper is a reasonable physical theory and is quite interesting. However, the overly introduced theory makes this article rather difficult to understand. I spent a lot of time trying to understand the physical theories presented in the text, but I still don't have a clear understanding yet. For example, Phase-TransitionLike Dynamics and Stability–Plasticity Dilemma are quite hard for AI researchers to understand.
- **RG formulation lacks formal grounding.** This method draws on the theory of RG. However, after researching RG, I found that the actual physical RG theory is quite distant from this paper. This paper feels more like a superficial covering. Specifically, due to the understanding difficulty, I personally think this paper is more like existing hierarchical memory mechanisms like MIRIX [1].

[1] MIRIX: Multi-Agent Memory System for LLM-Based Agents. Arxiv 2025.07.

---

> ### Author Rebuttal · Authors · 2026-03-30
>
> We thank you for the insightful review. Below, we clarify how RGMem differs from existing methods (e.g., MIRIX) in both static structure and dynamic evolution.
>
> #### **Response to W1 & W2: Theoretical Concepts**
>
> We here attempt to interpret the confusion in conjunction with physics terminology
>
> * **Stability–Plasticity Dilemma** is a classic problem in continual learning, referring to balancing long-term knowledge retention (stability) with adaptation to new information (plasticity).
> *  **Phase-Transition-Like Dynamics** that the macroscopic user profile does not change linearly with every interaction, but undergoes rapid restructuring only when accumulated evidence crosses a critical threshold.
>
> These two issues are exactly the weaknesses of traditional hierarchical memory, and RG theory provides a mathematical framework to address them.
>
>
> #### **Response to Q2: Fundamental Differences from Existing Works**
> Although RGMem appears similar to existing hierarchical memory mechanisms. However, RG theory guided four key design differences from philosophical viewpoint, by which RGMem can overcome the traditional limitations:
>
>
> **Static Structure**
>
> **Difference 1: Constrained Topology Converging to Fixed Points**
> Traditional graph-based methods allow topic nodes to drift and proliferate, leading to unstable long-term profiles.
> Inspired by RG flow converging to fixed points, we predefine 11 abstract nodes. Intermediate nodes may be generated, but all information is forced to converge to these fixed points via operator $R_{K3}$. This “locally free, globally convergent” structure ensures long-term macroscopic invariance.
>
> **Difference 2: Ontological Decoupling of Fast and Slow Variables**
> Traditional hierarchical memory mixes factual records and abstract preferences in the same compression path, causing either detail loss or high-level bloat.
> RG theory suggests separating **slow variables** (order parameters) from **fast variables**. We therefore separate Event nodes (fast variables, ensuring plasticity) and Abstract nodes (slow variables, ensuring stability). During coarse-graining ($R_{K2}$), fast-variable details are discarded while slow-variable features are propagated upward.
>
> **Dynamic Evolution**
>
> **Difference 3: Non-Linear Aggregation Preserving Contradictions**
> Traditional methods use similarity clustering or majority smoothing, which erases contradictory signals.
> In RG theory, coarse-graining must preserve critical fluctuations. Therefore, operator $R_{K3}$ extracts a tuple $(\Sigma, \Delta)$, where $\Sigma$ captures the dominant pattern and $\Delta$ preserves conflicting but important signals. This structured residual allows modeling user contradictions instead of averaging them away.
>
> **Difference 4: Critical-Threshold-Driven Phase-Transition Updates**
> Traditional systems use linear or periodic updates, making high-level profiles sensitive to noise and causing premature abstraction.
> RGMem introduces **critical-threshold updates**: macroscopic abstract nodes update only when accumulated low-level evidence exceeds a critical threshold $\theta_{inf}$, producing discrete structural shifts rather than continuous updates.
>
> Hence, in summary, the main difference lies in:
>
> * **RGMem emphasizes vertical evolution**: progressively abstracting relational patterns and high-level profiles from low-level events through thresholding and abstraction.
> * **LightMem uses process layering** for compression and offline updates.
> * **MIRIX uses type layering** for modular memory routing and retrieval.
>
> #### **Response to Q1: Figure 4 Interpretation**
> This explains the non-linear performance curve in Figure 4, which shows sensitivity to the evolution threshold $\theta_{inf}$ and indicates a critical point:
> * **Too low ($\theta_{inf} < 3$, subcritical):** overly sensitive to noise → unstable profiles.
> * **Too high ($\theta_{inf} > 3$, supercritical):** too slow to adapt → outdated profiles.
> * **Critical point ($\theta_{inf} = 3$):** optimal balance between **stability** and **plasticity**.
>
> This non-linear, threshold-controlled update mechanism is the key feature distinguishing RGMem from traditional linear or periodic updates.
>
>
> **Conclusion :** The four designs above, derived from RG theory at both static and dynamic levels, show that RG is not just a conceptual analogy, but a practical framework for addressing the stability–plasticity dilemma in long-term memory.

---

> > ### Author Rebuttal · Reviewer_brZr · 2026-04-02
> >
> > The author provides additional explanations for the terms, making it clearer. So I raised my scores. However, I still have concerns about the overly introduced theory. I would not mind if this paper gets rejected.

---

> > > ### Author Response · Authors · 2026-04-07
> > >
> > > ## LLM Agent Memory Lifecycle
> > > According to references [1-3],the LLM memory experiences a 5-module lifecycle:
> > > [M1.Formation]→[M2.Structuring]→[M3.Updating Triggers]→[M4.Consolidation]→[M5.Retrieval].While M1 & M5 are mature, ultra-long dynamic sequences still challenge the evolution modules (M2–M4).We next show how the RG-inspired framework resolves these issues.
> > > ## M2:Ontological Confusion & Clustering Limitation
> > > The hierarchical memory systems(e.g.,CAM[1],SGMem[4])rely on dynamic graphs & geometric clustering(KNN,semantic similarity)for macro-abstractions.Clustering captures only surface similarity, missing causal and logical relations.Over long dialogues,the lack of priors leads to chaotic macro-nodes and semantic drift.
> > > To solve this,our RG-inspired design clearly separates micro-details(fast var)from macro-traits(slow var).Instead of unconstrained clustering, we predefine semantic nodes as Fixed Points. All micro-events are mapped to these dimensions to form macro-traits.Such a strict mapping enables to avoid semantic drift and keeps the profile stable and interpretable.
> > > ### Table R1:RGMem vs.GraphRAG(a "clustering+summary" baseline fed with RGMem-L0 outputs)on 20% PersonaMem:
> > > |Method|Recall Facts|Track Pref.Evol.|Latest Pref|
> > > |-|-|-|-|
> > > |L0+GraphRAG|79.86|51.49|49.72|
> > > |RGMem|81.65|68.77|74.11|
> > > ### Analysis:
> > > i)Recall:GraphRAG retains basic facts
> > > ii)Macro Evolution:GraphRAG's accuracy drops by ~17-25% on dynamic tracking tasks.Similarity-based clustering lacks stable anchors,leading to progressive semantic drift over long dialogues
> > > iii)RGMem Advantage:By mapping micro-events to the predefined dimensions of Fixed Points,RGMem establishes a stable semantic coordinate system,improving tracking of profile changes.
> > > ## M3:Scale Mismatch in Updating Trigger
> > > Updating macro profiles is hard due to noise and delay.Existing methods usually use rigid external clocks:**high-frequent incremental updates**(e.g.,[5],per turn)inject noise, causing oscillation and wasted compute; **periodic global updates**(e.g.,[6],daily)lag behind changes and let local chats pollute profiles.
> > > Our RG's Critical Point module inspires local evidence density trigger.RGMem updates are event-driven:triggered only when micro-facts hit a threshold($\theta_{inf}=3$),allocating computation to structurally changing parts.
> > > ### Table R2:Cost & Acc.20% PersonaMem against high-freq($\theta_{inf}=1$)and periodic(scheduled global rewrite)updates.
> > > |Update Mech.|In Tok(k)|Out Tok(k)|Acc(%)|
> > > |-|-|-|-|
> > > |High-Freq|384.23|63.09|53.72|
> > > |Periodic|237.76|39.87|48.95|
> > > |RGMem|295.17|52.62|65.32|
> > > As shown in Table R2,compared to high-freq and periodic updates,RG's threshold-based trigger action significantly achieves the best Acc.performance.
> > > ### Table R3:Profile Sim.Convergence.Tracked cosine similarity $cos(\Sigma_{t},\Sigma_{t-1})$ of user profiles pre-/post-updates over 763 turns(1.0 = stable).
> > > |Update Mech.|T52|T149|T261|T356|T449|T567|T661|T763|
> > > |-|-|-|-|-|-|-|-|-|
> > > |High-Freq|0.00|0.61|0.91|0.65|0.37|0.54|0.64|0.66|
> > > |Periodic|0.00|0.65|0.72|0.61|0.78|0.66|0.81|0.71|
> > > |RGMem|0.00|0.69|0.75|0.91|0.93|0.89|0.92|0.90|
> > > ### Analysis:
> > > As shown in Table R3,high-Freq oscillates violently(no convergence).Periodic shows sawtooth reconstruction(recent chitchat pollutes stable traits).Differently,RGMem shows Phase-Transition-like Convergence:fast build,then stable,filtering noise.
> > > ## M4:"Over-smoothing" in Multi-level Compression
> > > Existing memory systems rely on LLMs for **Recursive Summarization** to compress long chats(e.g.,[7]).Yet,LLMs favor "Majority Vote",causing **Info smoothing**.During the **sudden preference reversals**,recent conflicting signals are "Averaged away" by long-time conversation history,causing sluggishness or "selective amnesia".
> > > Our RG's **Coarse-Graining** module demands preserving **Critical-Point Fluctuations**.That is to say,our operator abandons single summaries,forcing the extraction of tuple$(\Sigma,\Delta)$,to isolate new conflicting shifts $(\Delta)$ from dominant historical traits$(\Sigma)$.
> > > ### [Case Study]
> > > For the sentence:“History shows multiple strength training records.Latest turn shows one rehab exercise due to back injury.”Traditional summary treats injury as noise,yielding a false "heavy fitness" profile.Differently,our RGMem extracts $\Sigma$(Synergy)as "heavy fitness habit" AND $\Delta$(Tension)as "current rehab need," by which the over-smoothing can be well fixed.
> > > ### Table R4:Keep/remove $\Delta$ extraction
> > > |Aggregation Mech.|Latest Pref(%)|
> > > |-|-|
> > > |w/o $\Delta$ extraction|52.58|
> > > |with $\Sigma$ & $\Delta$|74.11|
> > > From the comparative results with/without extraction,we observe without system fails to capture the latest states while conflict exists,however,RG's extraction can fix it by preventing info loss.
> > > ## References
> > > [1] CAM...NeurIPS '25
> > >
> > > [2] From RAG to Memory...ICML '25
> > >
> > > [3] LIGHTMEM...ICLR '26
> > >
> > > [4] SGMEM...
> > >
> > > [5] Enabling personalized long-term interactions...
> > >
> > > [6] MemoryBank...AAAI '24
> > >
> > > [7] Iterresearch...ICLR '26

---

### Official Review · Reviewer_DdhH · 2026-03-13

**Soundness:** 3
**Presentation:** 3
**Significance:** 2
**Originality:** 3
**Overall Recommendation:** 4
**Confidence:** 3

**Summary:**

Inspired by the Renormalization Group (RG) theory, this paper effectively addresses the stability-plasticity dilemma inherent in current memory systems. The key contribution lies in its ability to deduce deeply-rooted user preferences from extensive historical interactions, while dynamically adapting to recent behavioral shifts, thereby generating high-quality and context-aware responses.

**Compliance With Llm Reviewing Policy:**

Affirmed.

**Final Justification:**

The rebuttal addressed my concerns, and I have maintained my positive assessment (Weak accept).

**Key Questions For Authors:**

1：Can it be empirically validated that this method is truly more efficient than the baselines when taking the entire end-to-end deduction mechanism into account?

2：Maintaining the memory graph is computationally expensive3The operators are instantiated via LLMs and prompt engineering; thus, if the LLM generates hallucinations during node abstraction, the error risks propagating and corrupting the overall memory graph.

**Limitations:**

yes

**Strengths And Weaknesses:**

Strength：

1：This work provides an in-depth analysis of the limitations inherent in current memory mechanisms. Addressing this critical dilemma, it proposes a comprehensive memory framework inspired by the Renormalization Group (RG) theory.

2：The core contribution lies in the integration of multi-level memory representations, enabling the dynamic and effective utilization of historical memories.

3：Furthermore, the proposed mechanism is solidly backed by both theoretical foundations and empirical validation.

Weekness：

1：Regarding the experimental evaluation, the paper only reports the token count of the final prompt sent to the LLM. When compared with the baselines, it remains to be verified whether the total token consumption, including the overhead incurred during the memory evolution and inference stages, is actually lower than that of alternative methods.

2：The operators are implemented using a combination of LLMs and prompts. If the LLM hallucinates during node-level abstraction, it will compromise the entire memory graph.

3：Maintaining the memory graph incurs substantial computational overhead.

---

> ### Author Rebuttal · Authors · 2026-03-30
>
> We thank the reviewer for recognizing our contributions and merits. Below, we address concerns regarding end-to-end efficiency, graph maintenance overhead, and hallucination propagation by adding new datasets and case studies.
>
> #### **(Response to Q1 & Q2) End-to-End Efficiency & Graph Maintenance Cost**
>
> As pointed out, evaluating only inference tokens is insufficient. Thus, we calculated the **average end-to-end cost per session** on PersonaMem (128k), encompassing dialogue processing, graph construction/maintenance, multi-scale retrieval, and generation.
>
> **Table 1: End-to-End Cost and Performance Comparison per Session on PersonaMem (128k)**
> | Method | Input Tok.(k) | Output Tok.(k) | Runtime(s) | API Calls | Acc.(%) |
> | :--- | :--- | :--- |:--- |:--- |:--- |
> | LangMem | 1128.17 | 134.51 | 2719.02 | 632.51 | 52.36 |
> | Mem0 | 1336.24 | 215.12 | 5507.65 | 1029.22 | 56.79 |
> | A-Mem | 1712.45 | 357.31 | 6312.21 | 1654.32 | 49.17 |
> | MemoryOS | 3489.71 | 480.61 | 9125.66 | 3278.42 | 54.23 |
> | **RGMem**| **1421.79** | **240.33** | **5441.02** | **978.54** | **63.87** |
>
> As shown in Table 1, our proposed RGMem fall in the mediate-level cost in general, however, it achieves a 7.08% accuracy gain compared to the best baseline Mem0 in Acc metric.
>
> Maintaining the multi-scale graph does not introduce significant overhead. This efficiency stems from our **non-linear, amortized update strategy**:
> 1) **Episodic Processing:** RGMem processes chunked topics rather than turn-by-turn, avoiding high-frequency trivial updates.
> 2) **Threshold-Driven Execution:**
> Our core operators ($(R_{K1}, R_{K2}, R_{K3})$)  are triggered only when accumulated microscopic evidence reaches predefined thresholds ($\theta_{inf}, \theta_{sum}$). The macroscopic graph structure remains stable most of the time and only undergoes localized reconstruction when necessary, ensuring long-term maintenance costs remain strictly controllable.
>
> #### **(Response to Q3) LLM Hallucinations and Error Propagation**
>
> RGMem's "coarse-graining" process inherently denoises and isolates hallucinations. As proven information-theoretically in Appendix B.1, majority-evidence aggregation improves the signal-to-noise ratio. To demonstrate how RGMem blocks a single hallucination from propagating upward, we trace a noisy fact through the operator flow ($R_{K1} \rightarrow R_{K2} \rightarrow R_{K3}$):
>
> **Step 1: L0 Fact Extraction.** The system extracts 6 Facts from a dialogue. Facts 1-5 (booking London flight/hotel/museum/musical, wanting afternoon tea) are correct. Fact 6 (*"User wants a romantic weekend in Paris"*) is an LLM hallucination.
>
> **Step 2: L1 Local KG Construction.** These are mounted under `Travel and Commute -> International Travel`.
> *   **Node A (London Trip)**: 3 edges (from Facts 1, 2, 3).
> *   **Node B (London Musical)** & **Node C (London Dining)**: 1 edge each (Facts 4, 5).
> *   **Node D (Paris Weekend)**: 1 edge (Fact 6, Hallucination).
>
> **Step 3:  $R_{K1}$** (Relation Inference, Threshold $\theta_{inf}=3$).
> *   **Node A Triggers:** With 3 edges, $R_{K1}$ aggregates Facts 1-3 into a stable relation $T_e^{(1)}$: *"User is actively organizing a summer trip to London, including flights, hotels, and museum visits."*
> *   **Nodes B, C, D Blocked:** Edge frequencies ($<3$) block $R_{K1}$. The hallucinated Fact 6 fails to form a stable relation and remains a raw string.
>
> **Step 4: $R_{K_2}$ (Node Abstraction, Threshold $\theta_{sum}=6$).**
> The `International Travel` node hits 6 pieces of evidence, triggering $R_{K_2}$.
> The input $I_v^{new}$ mixes Node A's reinforced $T_e^{(1)}$ with raw Facts 4, 5, and 6.
> Forced by the prompt to "synthesize commonalities across branches," the LLM finds strong semantic synergy (London, Culture) among A, B, and C.
> Fact 6 (Paris), an isolated outlier without $R_{K_1}$ support, is algorithmically filtered out as noise.
>
> * **Output Profile:** *"The approach favors comprehensive, full-scope planning, with a primary focus on deep exploration of cultural landmarks and immersive experiences of local culture."*
>
> **Step 5: $R_{K3}$** (Hierarchical Flow Propagation).
> The clean $R_{K2}$ profile propagates upwards to `Travel and Commute`:
> *   **Output Profile:** *"Culturally Driven Structured Travel Model: User exhibits a highly organized and culturally focused approach to travel."*
>
> **Conclusion:** The microscopic hallucination (Fact 6) is trapped at the bottom layer by $R_{K1}$'s threshold and filtered as noise during $R_{K2}$'s synergy extraction. It never corrupts the higher-level abstraction or the final retrieval context. We will integrate this analysis into the final version.

---

> > ### Author Rebuttal · Reviewer_DdhH · 2026-04-02
> >
> > Regarding the issue of end-to-end token consumption, the author provides detailed experimental data demonstrating that it also consumes fewer tokens than other baselines in end-to-end tasks. Meanwhile, strict proofs and explanations are given regarding whether prompt-based mechanisms may induce hallucinations that damage graph structures, which addresses my questions.

---

### Decision · Program_Chairs · 2026-04-30

**Decision:**

Accept (regular)

**Comment:**

This paper learns user preferences from long-conversational history. The framework inspired by Renormalization Group (RG) separates fast-changing evidence from slow-changing preferences. Experiments show that this framing leads to better performance.

Reviewers liked the RG framing and empirical results. The in-depth analysis of the limitations of existing memory-systems was also appreciated.

Main weakness that were raised were:

1. Memory graph maintenance adds an overhead and is prone to LLM hallucination. Authors clarify that the coarse-graining process in RGMem is robust to hallucination. This intuitively makes sense as you take majority evidence in account.

2. It is unclear if the total token consumption is less than existing methods. Authors have provided results showing that the total token consumption is intermediate amongst a range of methods, while it gives a 7% improvement in accuracy.

3. RG theory in the paper is hard to read and it is unclear what advantages it incurs. The existing method seems similar to current memory techniques like recursive summarization, and so it is unclear if RG theory is even relevant. Authors have provided explanations on how RG theory guides the solution, however, the end-result does seem similar to existing memory methods. Further, authors have provided theoretical proofs to support their argument. I think it is not uncommon for theoretically-inspired methods to find results that are similar to those that have been arrived by more empirical arguments. It is possible that the conceptual framing leads to further improvements, or helps explain why existing methods work.

Overall, I lean towards a weak accept on this one.